Subject Areas:
systems biology/chemical biology/computer modelling and simulation

Keywords:
chemical kinetics, differential equations, game theory, protein aggregation

Author for correspondence:
Preetam Ghosh
e-mail: pghosh@vcu.edu

# A game-theoretic approach to deciphering the dynamics of amyloid-β aggregation along competing pathways

Preetam Ghosh[1], Pratip Rana[1], Vijayaraghavan Rangachari[2], Jhinuk Saha[2], Edward Steen[3] and Ashwin Vaidya[3]

[1]Department of Computer Science, Virginia Commonwealth University, Richmond, VA 23220, USA
[2]Department of Chemistry and Biochemistry, School of Mathematics and Natural Sciences, University of Southern Mississippi, Hattiesburg, MS 39406, USA
[3]Department of Mathematical Science, Montclair State University, Montclair, NJ 07043, USA

PG, 0000-0003-3880-5886

Aggregation of amyloid-β (Aβ) peptides is a significant event that underpins Alzheimer's disease (AD). Aβ aggregates, especially the low-molecular weight oligomers, are the primary toxic agents in AD pathogenesis. Therefore, there is increasing interest in understanding their formation and behaviour. In this paper, we use our previously established results on heterotypic interactions between Aβ and fatty acids (FAs) to investigate off-pathway aggregation under the control of FA concentrations to develop a mathematical framework that captures the mechanism. Our framework to define and simulate the competing on- and off-pathways of Aβ aggregation is based on the principles of game theory. Together with detailed simulations and biophysical experiments, our models describe the dynamics involved in the mechanisms of Aβ aggregation in the presence of FAs to adopt multiple pathways. Specifically, our reduced-order computations indicate that the emergence of off- or on-pathway aggregates are tightly controlled by a narrow set of rate constants, and one could alter such parameters to populate a particular oligomeric species. These models agree with the detailed simulations and experimental data on using FA as a heterotypic partner to modulate the temporal parameters. Predicting spatio-temporal landscape along competing pathways for a given heterotypic partner such as lipids is a first step towards simulating scenarios in which the generation of specific 'conformer strains' of Aβ could be predicted. This approach could be significant in deciphering

the mechanisms of amyloid aggregation and strain generation, which are ubiquitously observed in many neurodegenerative diseases.

## 1. Introduction

Aggregation of the protein amyloid-$\beta$ (A$\beta$) is one of the central processes in the aetiology of Alzheimer's disease (AD). Generated by the proteolytic processing of amyloid precursor protein (APP), A$\beta$ peptides (A$\beta$40 or A$\beta$42) spontaneously aggregate to form insoluble fibrils that deposit as senile plaques in the AD brain. During aggregation, the low-molecular weight oligomers formed are known to be the primary toxic species responsible for synaptic dysfunction and neuronal loss [1–6]. An increasing number of reports indicate that structural polymorphism and heterogeneity within the aggregates could contribute to clinical phenotypes observed among AD patients [7,8]. Therefore, over the last decade, significant efforts are focused on understanding the biophysical and biochemical aspects of aggregation as well as the molecular understanding of the aggregates.

A$\beta$ aggregation follows a nucleation-dependent, sigmoidal growth kinetics involving a key rate-limiting event of nucleus or nuclei formation [9–13]. Since the nucleation plays an important role in determining the morphology of the fibrils formed, the dynamics associated with reactions leading up to nucleation are critical determinants of aggregation. In this regard, the sensitivity of A$\beta$ to environmental factors and many interacting partners due to its intrinsic disorder and amphipathic nature [14–18] play a key role in A$\beta$ adopting multiple pathways depending on the aggregation conditions. An important ramification of this is that the oligomers may not be obligate intermediates of fibril formation but those with distinct conformations can be formed along alternative aggregation pathways (off-pathways) [13,19–23]. This is significant because such interactions, depending on the structure of the oligomer, determine the morphology of the aggregates formed and consequently, the toxicity and phenotypes.

Therefore, it is imperative to gain an understanding of how physiological interacting partners of A$\beta$ affect its aggregation dynamics. Being generated from the membrane-spanning domain of the APP, A$\beta$ displays synchronous and perpetual interaction with membrane lipids [24–30]. Interfaces of lipids and fatty acids (FAs) are also abundant in both cerebral vasculature and cerebral spinal fluid (CSF) [31,32]. Previous reports have established that phase transitions of surfactants and membrane lipids modulate A$\beta$ aggregation in a concentration-dependent manner to generate aggregates by an alternative, off-pathway from the canonical fibril formation, on-pathway [13,16,20,33–37]. Specifically, in micellar lipids, low-molecular weight oligomers were generated along off-pathway in the presence of fatty acid near and above their respective critical micelle concentrations (CMC) (pseudo-micellar and micellar, respectively) and not below CMC (non-micellar) which augmented the fibril formation in the on-pathway [16,34,38].

The modulation of aggregation by heterotypic interactions between A$\beta$ and lipids posit the question of what spatio-temporal parameters govern the modulatory dynamics, and whether one could simulate the temporal emergence and disappearance of aggregates as a function of heterotypic A$\beta$-lipid interactions. In this work, we have approached to answer these questions using a competition-based (built qualitatively upon the idea of game theory) approach to determine the dynamics in the temporal evolution of A$\beta$ aggregates along the pathways influenced by fatty acid surfactants ($L$). Our rationale for such an approach is that the stochasticity and the often exclusive pathways of A$\beta$ aggregation present 'win or loss' scenarios with respect to pathway adoption, tightly governed by the concentration and phase transitions of $L$. The mathematical analysis of this problem was taken up in two layers, one feeding into the other. The first is a six species, coarse-grained, reduced-order model (ROM), while the second is a more detailed model called ensemble kinetic simulation (EKS), which captures the temporal kinetics of reactions at the atomistic scale (considered as point particles). The ROM approach lends itself to a detailed analysis in a manner that cannot be performed in high-resolution models as we have shown before [34,39]. Phenomenologically inspired by the biophysical framework, and using toy models, ROM provides insights into the dynamics of mechanism that are previously unknown. In addition, the outcomes of the ROM analysis provide the appropriate cues to investigate the mechanism deeper with the EKS models. These models are partly validated by bulk kinetic and thermodynamic features using biophysical experiments. The simulations, supported by biophysical analyses, provide a temporal contour map along competing pathways, and present a unique perspective on otherwise unknown evolution of aggregates along multiple pathways.

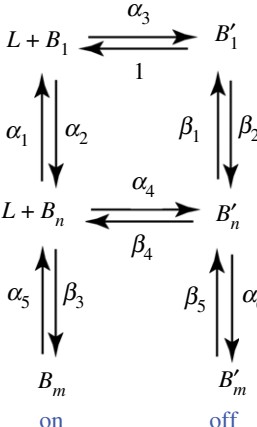

**Figure 1.** Schematic of on- and off-pathway aggregation model based on the six-species reaction scheme described earlier.

# 2. Material and methods

## 2.1. Reduced-order kinetic modelling

The model presented here consists of a reduced order, comprising only six species of A$\beta$ that interact with the fatty acid surfactant, $L$. Even with just six species, there are infinitely many rate regimes, most of which would be physically inconsequential. Thus, only physically meaningful rate regimes suggested from experiments and our previous studies [34,39] were chosen, and key parameters were varied to understand the dynamics. Specifically, two models were considered: (i) the base model, where the forward rate constants to back constants were taken to be 1000, and (ii) a second 'pathological' model, where the forward and backward reactions are taken to be identical. The second model has no known physical basis; however, it can be considered as a sort of parameter sensitivity study and an extreme case when the physiological process breaks down.

A schematic of such a model is presented below (see also figure 1). In this model, A$\beta$ monomers react with the pseudo-micellar fatty acid surfactants, $L$ to modulate the formation of on- or off-pathway aggregates. The system of chemical reactions in our model consists of the following:

$$A_1 + L \xleftrightarrow[k_1^-]{k_1^+} A_1',$$

$$nA_1 \xleftrightarrow[k_2^-]{k_2^+} A_n,$$

$$nA_1' \xleftrightarrow[k_3^-]{k_3^+} A_n',$$

$$A_n + L \xleftrightarrow[k_4^-]{k_4^+} A_n',$$

$$\frac{m}{n} A_n \xleftrightarrow[k_5^-]{k_5^+} A_m$$

and

$$\frac{m}{n} A_n' \xleftrightarrow[k_6^-]{k_6^+} A_m'.$$

The non-prime species, $A_1$, $A_n$ and $A_m$ represent on-pathway A$\beta$ monomers ($A_1$) and oligomers ($A_n$ and $A_m$ where $m$ is an integral multiple of $n$); whereas the prime species, $A_1'$, $A_n'$ and $A_m'$, are the corresponding off-pathway species which are generated through a reaction with the pseudo-micellar surfactant, $L$. The rate constants $k_i^{\pm}$ ($i = 1$–6) are indicated in the reaction schematic above where the '+' represents a forward rate and '−', a backward rate. These reactions were formulated based on experimental evidence demonstrated earlier [40]. In the computations to follow, for each species, $n = 4$ and $m = 20$ unless otherwise specified, which denotes the order of oligomer [33]. The $n$, $m$ values in the computations were kept low to minimize computational time. This is also because only significant qualitative features in the system were being sought by ROM, and a more fine-grained approach by

EKS modelling provides atomistic temporal analyses using the output from ROM. However, it must be noted that the key results of the study were examined for different values of $n$ and $m$ to ensure qualitative similarities and with no loss of generality as shown previously [34].

The reaction scheme was used to develop the corresponding kinetic model comprising a system of six nonlinear differential equations. This system was then put into non-dimensional form. Using $A_0$ as the characteristic concentration of monomers and $1/k_1^-$ the characteristic time, the dimensionless species are defined as follows:

$$B_1 = \frac{A_1}{A_0}; \quad B_n = \frac{A_n}{A_0}; \quad B_m = \frac{A_m}{A_0}; \quad B_1' = \frac{A'_1}{A_0}; \quad B_n' = \frac{A'_n}{A_0}; \quad B_m' = \frac{A'_m}{A_0}.$$

The reaction constants are similarly defined as follows:

$$\alpha_1 = \frac{k_2^-}{k_1^-}; \quad \alpha_2 = \frac{k_2^+ A_0^{n-1}}{k_1^-}; \quad \alpha_3 = \frac{k_1^+ L}{k_1^-}; \quad \alpha_4 = \frac{k_4^+ L}{k_1^-}; \quad \alpha_5 = \frac{k_5^-}{k_1^-}; \quad \alpha_6 = \frac{k_6^+ A_0^{(m/n)-1}}{k_1^-}$$

and

$$\beta_1 = \frac{k_3^-}{k_1^-}; \quad \beta_2 = \frac{k_3^+ A_0^{n-1}}{k_1^-}; \quad \beta_3 = \frac{k_5^+ A_0^{(m/n)-1}}{k_1^-}; \quad \beta_4 = \frac{k_4^-}{k_1^-}; \quad \beta_5 = \frac{k_6^-}{k_1^-}.$$

Note that both $\alpha_3$ and $\alpha_4$ have a factor $L$ which is responsible for off-pathway aggregation. These two parameters serve as the *bridge variables* between on- and off-pathway species. Using the law of mass action kinetics, the dimensionless system of differential equations was formulated as follows:

$$\frac{dB_1}{ds} = n\alpha_1 B_n - n\alpha_2 B_1^n + B_1' - \alpha_3 B_1, \tag{2.1}$$

$$\frac{dB_1'}{ds} = n\beta_1 B_n' - n\beta_2 B_1'^n + \alpha_3 B_1 - B_1', \tag{2.2}$$

$$\frac{dB_n}{ds} = \alpha_2 B_1^n - \alpha_1 B_n + \frac{m}{n}\alpha_5 B_m + \beta_4 B_n' - \alpha_4 B_n - \frac{m}{n}\beta_3 B_n^{m/n}, \tag{2.3}$$

$$\frac{dB_n'}{ds} = \beta_2 B_1'^n - \beta_1 B_n' + \alpha_4 B_n + \frac{m}{n}\beta_5 B_m' - \frac{m}{n}\alpha_6 B_n'^{m/n} - \beta_4 B_n', \tag{2.4}$$

$$\frac{dB_m}{ds} = \beta_3 B_n^{m/n} - \alpha_5 B_m \tag{2.5}$$

and

$$\frac{dB_m'}{ds} = \alpha_6 B_n'^{m/n} - \beta_5 B_m'. \tag{2.6}$$

As stated earlier, primarily two models referred to as the *Base Model* and *Model 2* were analysed, which are distinguished by the choice of fixed parameter values; i.e. the rate constant ratios in the pure on- and off-pathways. In the *Base Model*, all forward rates ($\alpha_1$, $\alpha_2$, $\alpha_5$, $\alpha_6$) and all backward rates ($\beta_1$, $\beta_2$, $\beta_3$, $\beta_4$, $\beta_5$) were set to 1 and 0.001 based on previous mathematical models and experimental data [39,40]. In the context of the *Base Model*, a forward rate is defined as one that converts a smaller oligomer into a larger aggregate, and backward being the reverse process. It must be noted that since ROM is a bulk averaged model, precise one-to-one mapping of its rate constants to that of the detailed EKS model is neither practical nor meaningful. In *Model 2*, all forward and backward rates were set to unity. *Ode 45* solver (Matlab) was used for our numerical computations.

A convenient approach to the problem would be to analyse the model equations (2.1)–(2.6) from a game-theoretic point of view. Such an approach warrants finding the conditions under which the triplet ($B_1$, $B_n$, $B_m$) are greater, less or equal to ($B_1'$, $B_n'$, $B_m'$) respectively; equality would indicate the *Nash equilibrium*. A similar game-theoretic treatment was applied to a simpler system in our earlier work on multiple-pathway protein aggregation [34], and also by others on various biochemical systems [41–43]. In the context of amyloid protein aggregation, the current model system shows the emergence of new states discussed in detail in §3.1.3, which have previously not been observed and lead to new experimental questions about dominant chemical reaction fluxes in competing systems.

## 2.2. Ensemble kinetic simulation

Detailed insights into the switching behaviour between on- and off-pathways were formulated by a combined off–on-pathway EKS model. EKS model has previously been applied for $A\beta$ aggregation system [11,12,34,39,44–46]. In this paper, we have extended our previous work by adding switching

reactions considering off-to-on and on-to-off oligomer conversion. It has to be borne in mind that the switching reactions only take effect from perturbation events such as changes in the concentrations of pseudo-micellar fatty acid, $L$ [16].

In the EKS model, a set of reactions was considered to represent the on-pathway, off-pathway and their switching, and the flux for each reaction was computed. The system of differential equations of each species present in the reaction system were identified and solved using the *ODE 23s* solver (Matlab). The following is the reaction scheme considered (corresponding differential equations are presented in Appendix C):

I. *Reactions of on-pathway: (considering $A_{12}$ as F)*:

$$A_1 + A_i \underset{k_{nu\_}}{\overset{k_{nu}}{\longleftrightarrow}} A_{i+1}; \quad \forall i \in \{1,2,\ldots,11\}$$

and

$$F + A_i \underset{k_{el\_}}{\overset{k_{el}}{\longleftrightarrow}} F; \quad \forall i \in \{1,2,\ldots,11\}.$$

II. *Reactions of off-pathway model*:

$$4\,A_1 + L \underset{k_{con\_}}{\overset{k_{con}}{\longleftrightarrow}} A_4',$$

$$A_i' + A_1 \underset{k_{nuf\_}}{\overset{k_{nuf}}{\longleftrightarrow}} A_{i+1}'; \quad \forall i \in \{4,\ldots,11\},$$

$$A_{12}' + A_i' \underset{k_{el1f\_}}{\overset{k_{el1f}}{\longleftrightarrow}} F_1'; \quad \forall i \in \{4,\ldots,11\},$$

$$F_1' + F_i' \underset{k_{el2f\_}}{\overset{k_{el2f}}{\longleftarrow}} F_{i+1}'; \quad \forall i \in \{1,\ldots,3\}$$

$$\text{and} \quad F_4' \underset{k_{fagf\_}}{\overset{k_{fagf}}{\longleftrightarrow}} 4F_1''.$$

III. *On-to-off switching reaction*:

$$A_i' \underset{k_{swi\_}}{\overset{k_{swi}}{\longleftrightarrow}} A_i.$$

The corresponding flux for the reactions is given as follows:

I. *On-pathway reactions flux*:

$$H_i = k_{nu}[A_i][A_1] - k_{nu\_}[A_{i+1}]; \quad \forall i \in \{1,2,\ldots,11\}$$
$$I_i = k_{el}[A_i][F] - k_{el\_}[F]; \quad \forall i \in \{1,2,\ldots,11\}.$$

II. *Off-pathway reactions flux*:

$$G_1' = k_{con}[A_1]^4[L] - k_{con\_}[A_4'],$$
$$H_i' = k_{nuf}[A_i'][A_1] - k_{nuf\_}[A_{1+i}']; \quad \forall i \in \{4,\ldots,11\},$$
$$I_i' = k_{el1f}[A_i'][A_{12}'] - k_{el1\_}[F_1']; \quad \forall i \in \{4,\ldots,11\},$$
$$P_i' = k_{el2f}[F_i'][F_1'] - k_{el2\_}[F_{i+1}']; \quad \forall i \in \{1,\ldots,3\}$$

and
$$R_1' = k_{fagf}[F_4'] - k_{fagf\_}[F_1'']^4.$$

III. *Switching flux*:

$$S_i = k_{swi}[A_i'] - k_{swi\_}[A_i].$$

Here, and in Appendix C, $A_i$ denotes an on-pathway *i*-mer, $A_i'$ denotes an off-pathway *i*-mer, $L$ denotes pseudo-micellar surfactants, $F$ denotes post-nucleation on-pathway aggregates (here $A_{12}$ is considered equivalent to $F$ which corresponds to an on-pathway nucleus of 12mer as previously reported [40]; $F$ for the sake of simplicity), $F_i'$ is an off-pathway oligomer, *signal* is the total thioflavin-T (ThT) fluorescence intensity which is expressed as the sum of the on-pathway ThT signal (signal$_{on}$) and the off-pathway ThT signal (signal$_{off}$) (as shown in Appendix B; this uses an arbitrary mapping constant to map the total oligomer concentration to the experimentally observed ThT signal intensity).

Note that in the EKS models, we consider the most general case where there can be switching between any on- or off-pathway oligomer of size $A_1$ to $A_{11}$. Similarly, smaller off-pathway oligomers from $A'_{16}-A'_{23}$ were considered as $F'_1$ and larger off-pathway oligomers were considered as $F'_i (i = 1, \ldots, 4)$ while a dissociation of $F'_4$ was considered to lead to the formation of $F''_1$, which is a kinetically trapped off-pathway oligomer that does not aggregate further. The existence of such on- and off-pathway oligomers and the validity of our combined on- and off-pathway model (barring the switching reaction) have already been established in earlier work [34,47].

## 2.3. Biophysical analysis

Synthetic, wild-type A$\beta$42 procured from both Peptide 2.0 and Dr Chaterjee's laboratory at the University of Mississippi was used in this study. ThT, sodium dodecyl sulfate (SDS) and lauric acid (C 12:0) was purchased from Sigma–Aldrich (St Louis, MO). Monoclonal Ab9 or Ab5 antibodies were obtained from the University of Florida Center for Translational Research in Neurodegenerative Diseases.

### 2.3.1. Protein preparations

*Preparation of A$\beta$42 monomers:* A$\beta$42 peptide (1–1.5 mg) that was kept desiccated at $-20°C$ was dissolved in 50 mM NaOH and was allowed to stand at room temperature for up to 45 min. The dissolved peptide was then fractionated on a Superdex-75 HR 10/30 size exclusion chromatography (SEC) column (GE Life Sciences) on a BIORAD FPLC system that was pre-equilibrated with 20 mM Tris at pH 8.00, to separate any preformed aggregates as previously reported [41]. Fractions were collected at a flow rate of 0.5 ml min$^{-1}$ and stored at 4°C and were used within 24 h to avoid reaggregation. The concentration of the monomeric fractions was calculated using a Cary 50 UV–Vis spectrophotometer (Varian Inc.). The molar extinction coefficient of 1450 cm$^{-1}$ M$^{-1}$ at 276 nm was used (www.expasy.org).

*On- and off-pathway aggregation reactions:* On-pathway aggregation was initiated with 40 µM monomeric A$\beta$42 in 20 mM Tris–HCl, 50 mM NaCl at pH 8.0 incubated under quiescent conditions at 37°C with 0.01% NaN$_3$. Off-pathway reactions were initiated using 25 µM monomeric A$\beta$42 in the same buffer incubated with 50 mM NaCl and 5 mM sodium laurate (C12 FA) in 20 mM Tris, pH 8.00, as reported previously [16,34].

### 2.3.2. ThT fluorescence aggregation assay

For on-to-off-pathway switching reactions, to 150 µl, 50 µM A$\beta$ reactions incubated in buffer alone, a 50 µM ThT solution in the same buffer was added and fluorescence emission ($\lambda = 482$ nm) was collected using microplate reader (BioTek Synergy Microplate Reader) at 37°C using an excitation at 452 nm. A 5 mM sodium laurate (C12 fatty acid) sample pre-equilibrated with 50 µM ThT was added to the reactions at 3, 8 and 24 h to initiate switching of pathways. The data were collected at 10 min time intervals. For off-to-on-pathway switching reactions, the 150 µl, 50 µM A$\beta$ reactions pre-incubated in the presence of 5 mM sodium laurate were diluted 5- or 10-fold at 5 and 10 h using buffered 50 µM A$\beta$ monomers and 50 µM ThT such that only the fatty acid concentration is dropped below its CMC. Appropriate blank reactions were monitored simultaneously and were corrected before data processing.

### 2.3.3. SDS–PAGE and immunoblotting

Aliquots of the reactions were mixed with sample buffer comprising 1% SDS (1× Laemmli sample buffer) and loaded on a precast 4–12% Bio-Rad gel. For calibration, pre-stained molecular weight markers (Invitrogen Inc.) were used. The gels were then electroblotted on 0.45 µm nitrocellulose membrane (GE Life Sciences). The blots were then heated in the microwave for 1 min and were blocked with 5% non-fat dry milk solution with 1% Tween-20 in PBS for 1.5 h. Subsequently, the blots were probed with monoclonal antibody Ab5 or Ab9 (1 : 1000–1 : 2500 dilutions) which bind to residue 1–16 of A$\beta$. Anti-mouse horseradish peroxide was added to the blot and the blot was developed with ECL reagent (Thermo Fisher Scientific) and imaged with a Bio-Rad Gel Doc system.

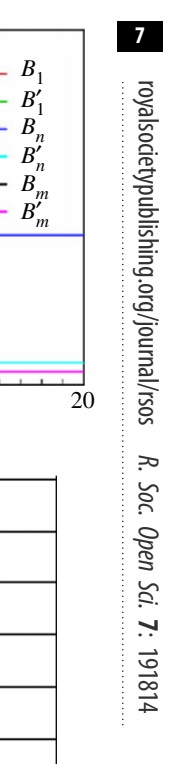

**Figure 2.** Panels (*a*) and (*b*) show sample solutions of the *Base Model* and *Model 2* corresponding to equations (2.1)–(2.6) for the ROM. The different colours in both panels correspond to the evolution of the six different species, indicated in the figure legend. Panel (*c*) depicts a table showing the equilibrium time as a function of the ratio of backward to forward rates for the *Base Model*. Clearly, as the ratio of backward to forward rate constants increase, as in pathological cases, the time to equilibrium decreases.

| backward/forward ratio | equilibrium time |
|:---:|:---:|
| 0.0001 | 5190 |
| 0.001 | 2200 |
| 0.01 | 595 |
| 0.1 | 105 |
| 1 | 20 |

## 3. Results

### 3.1. Reduced-order model indicates switching behaviour between pathways is dictated by the dynamics of equilibrium stability and bridging

#### 3.1.1. Steady states

In order to study the stability of the system, bulk rate constants obtained from experiments were used to determine steady state, or concentration at a given time-point [39]. One is especially interested in the non-zero terminal states of each species. Numerical computations indicate that concentration of species continues to change over time for our models, heading towards the steady state. However, in all cases these changes were within 0.1% of previous levels for $t$ greater than some critical time, which was considered acceptable as equilibrium. The equilibrium values were also confirmed through Matlab's *fsolve* function. In all ROM computations discussed in this paper, the initial conditions were taken to be $B_1(0) = 1$ with all other species set initially at zero. As seen from figure 2*a* and *b*, as time increased, the concentration levels exhibited asymptotic behaviour and each species eventually achieved equilibrium. The time to reach this steady state was sensitive to the choice of rate constants; the *Base Model* took longer to reach steady state than the *Model 2*. Also, in all cases analysed the fibril concentrations $B_m$ and $B'_m$ took the longest to reach equilibrium.

Due to low forward rates, the concentration size of $B_1$ stayed high and stable throughout, but large percentage changes in the concentration of $B'_m$ were observed periodically. Analysis of the concentration patterns of both $B_1$ and $B'_m$ over this period revealed that their growth and decline patterns are a reciprocal image of one another: periods of increase in $B_1$ were accompanied by decline in $B'_m$. From the equilibrium analysis of these models, it was discovered that when the ratio of backward to forward rates is close to 1, the model settles at equilibrium more quickly than when the ratio is large (figure 2*c*). A power-law regression indicates that time to equilibrium (indicated by $t_{eq}$) varies with the ratio of forward to backward rates ($r$) according to $t_{eq} \propto r^{-0.615}$.

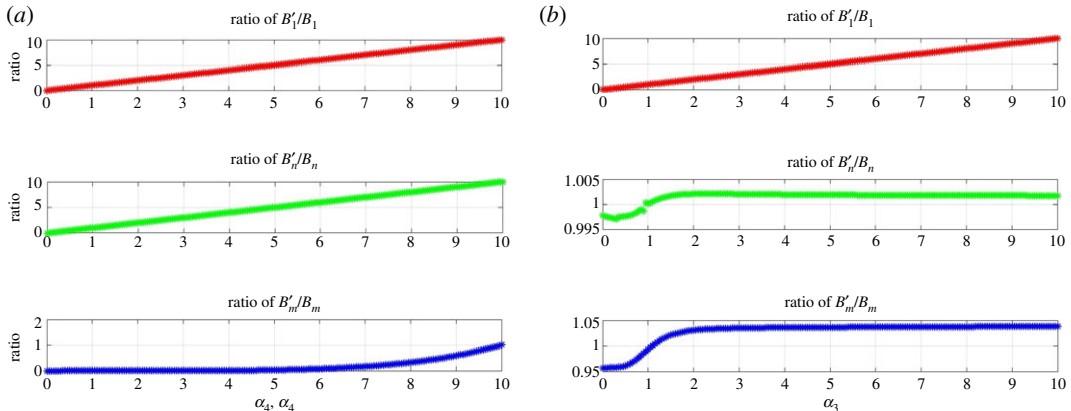

**Figure 3.** Concentration ratios of like-species between the two pathways as a function of $\alpha_3$ and $\alpha_4$ in the *Base Model*. Panel (*a*) shows these ratios as a function of both the bridge parameters while in panel (*b*) $\alpha_3$ is varied while holding $\alpha_4$ fixed. This figure shows the impact of the bridge parameters upon specific oligomers in the reaction pathways, indicating that the bridges between larger oligomers play a more significant role in the pathway dynamics and competition.

The impact of varying $n$ and $m$ in both models was also investigated. In these cases, increasing $n$ and $m$ increased $t_{eq}$ but yielded similar qualitative results, some of which are also discussed in Appendices A and B. It appears that the larger the oligomer size, the higher the power of the nonlinear terms in the governing equations, the greater the potential for over- and under-shoot as the model evolves over time. Thus, it takes longer to achieve equilibrium. However, *Model 2* does not show an increase in $t_{eq}$ as noted earlier.

### 3.1.2. Bridge parameters

The key parameters in our model are $\alpha_3$ and $\alpha_4$ which are referred to as 'bridge' or control parameters, and they govern the reaction dynamics between on- and off-pathway. The effect of varying both on species formation was verified while holding all other reaction rates constant (figure 3). When increasing both $\alpha_3$ and $\alpha_4$, a direct increase in the ratio of off-pathway species to on-pathway species was observed. Since $B'_m/B_m$ is not directly governed by the bridge variables, it was slower to react to changes along the bridges, but eventually exhibited what appears to be exponential growth at higher values of the bridge variables (figure 3*a*). This is probably due to the fact that $B'_m$ formation is dependent upon $\alpha_3$ and $\alpha_4$, so that increasing $\alpha_3$ and $\alpha_4$ eventually impacts $B'_m$.

Figure 3*a* and *b* underscores the importance of the bridge variables. Interestingly, if $\alpha_4$ was left unchanged and $\alpha_3$ was increased, there was limited flow-through from $B'_1$ to $B'_n$ and $B'_m$: their ratios to the non-prime species increased slightly above unity, but ceased to grow from there on even as $\alpha_3$ continued to increase. Therefore, this suggests that the bridge reaction $B_n \leftrightarrow B'_n$ is critical in the formation of the larger oligomers, i.e. the $n$ and $m$ species. The ROM modelling, therefore, reveals that bridges between larger oligomers are more significant than the ones across monomers in terms of promoting off-pathway fibril formation. Additional tests were performed to verify conditions for any species to outperform others by appropriate choice of the rate constants. Forcing $B_1$ to outperform, for instance, is just a matter of reducing or shutting off all the forward reactions. For species further down the reaction-network, forward reactions were required to increase to get the desired out-performance. In the case of $B'_m$, out-performance of this species was obtained in absolute terms by increasing the forward reaction rates $\alpha_3$, $\beta_2$ and $\alpha_6$ by an order $10^4$. Out-performance by $B_m$ could also be achieved in a similar manner. Such an exercise can be significant in helping to identify pathogenic aggregates and shows the robustness of the network under standard reaction rates.

### 3.1.3. A 'game-theoretic' approach to understanding pathways

Figure 4*d* provides a schematic of the four equilibrium pathways that our model can achieve, each sensitive to the choices of parameters $\alpha_3$ and $\alpha_4$. In figure 4*a*, the first schematic highlighted in red is strictly on-pathway, where the non-prime species 'win'. The next highlighted in blue is strictly off-pathway, where all off-pathway species wins. The paths indicated in yellow and green are a mixture of on/off-pathway. Figure 4*d* depicts the network graph corresponding to each of the phases. Computations were conducted

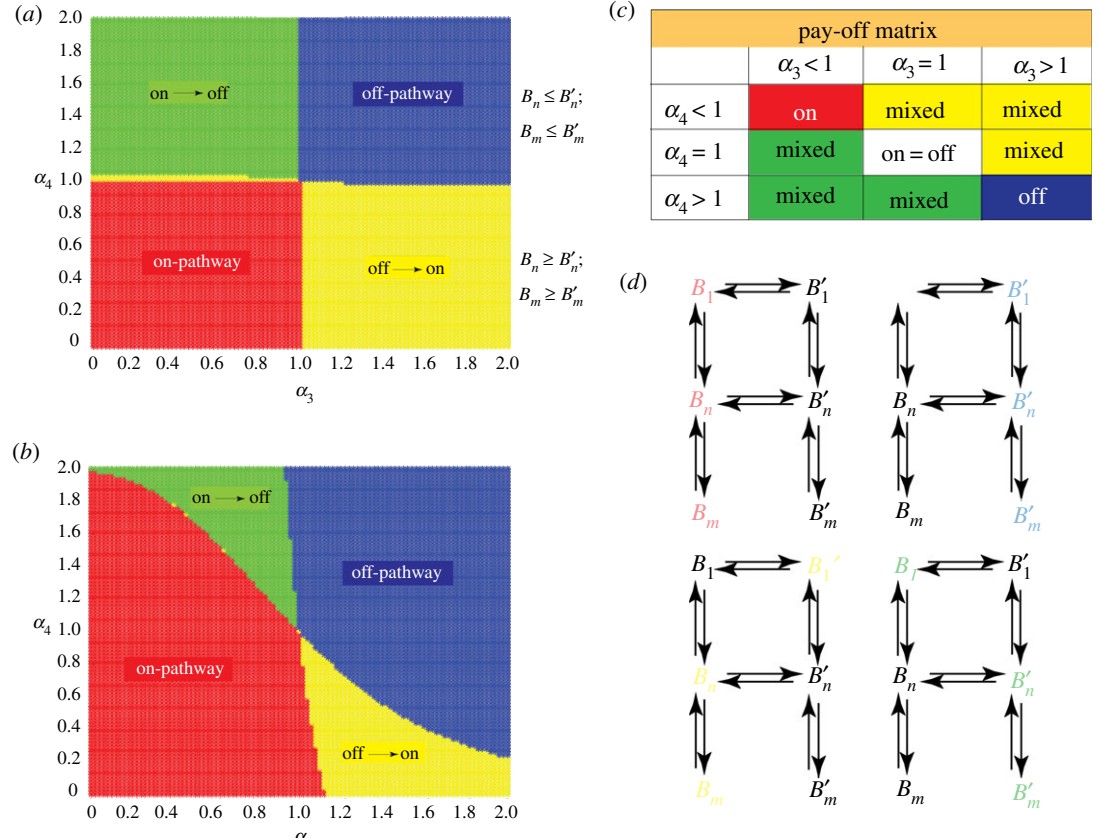

**Figure 4.** Aggregation pathways as a function of $\alpha_3$ and $\alpha_4$. Panels (a) and (b) depict a contour plot of the dominant species as a function of the bridge parameters for the *Base Model* and *Model 2*, respectively. Panel (c) shows the pay-off matrix for panel (a) depicting the various conditions for domination. Panel (d) depicts the pathway diagram indicating the dominant sub-path for specific choices of bridge parameters, corresponding to panels (a) and (b).

by varying $\alpha_3$ and $\alpha_4$ between 0 and 2 in increments of 0.02, resulting in 10 000 discrete points. Figure 4a and b shows a phase diagram for the *Base Model* and *Model 2*, respectively.

For the *Base Model*, $(\alpha_3, \alpha_4) = (1, 1)$; is a critical point at which the concentrations of on- and off-pathway species are equal. As $\alpha_3$ and $\alpha_4$ were varied, dominance of one set of species or pathways over another emerged. Notable too is the fact that the boundaries between the different equilibrium states were almost linear: the line $\alpha_4 = 1$ determines the switching between on- and off-pathway dominance of $n$ and $m$ species, and the line $\alpha_3 = 1$ determines the switching between on- and off-pathway dominance of monomers. The table in figure 4c shows the equilibrium states as a function of $\alpha_3$ and $\alpha_4$ in the form of a pay-off-like matrix. The *Nash equilibrium* lies at the point where on-pathway species concentrations are equal to off-pathway species concentrations.

A similar computation was performed for *Model 2* (figure 4b). Here too, $(\alpha_3, \alpha_4) = (1, 1)$ was a critical point; however, unlike in the *Base Model*, it does not strictly define out-performance of $B_1$ over $B'_1$ and vice versa; still $B_1$ outperforming $B'_1$ was seen for $\alpha_3 > 1$ and low $\alpha_4$, and $B'_1$ outperforming $B_1$ for $\alpha_3 < 1$ and high $\alpha_4$. The major difference is that the red and blue regions representing the only on-pathway and only off-pathway, respectively, increased at the expense of green and yellow (mixed pathways).

More importantly, the range of points over which on-pathway wins got bigger when backward rates were on par with forward rates. For $\alpha_3 < 1$, by increasing $\beta_1$ and $\beta_5$, $B_1 \rightarrow B'_1$ bridge towards off-pathway aggregation was effectively shut off, hence the increase in red in the upper left of figure 4b. The reverse happened in the lower right for $\alpha_4 < 1$ as we observed greater off-pathway aggregation up to a point. Despite increasing $\alpha_3$ above 2, upon reducing $\alpha_4$, a thin band of dominance of $B_n$ and $B_m$ over their respective primed off-pathway species was continued to be observed. Once again, this shows that the $B_n \rightarrow B'_n$ bridge is more critical for off-pathway aggregation of $n$ and $m$ species than the monomer-bridge. If $\alpha_4$ was reduced, an on-pathway dominance of $n$ and $m$ species even for high $\alpha_3$ was still obtained. Thus, it is difficult to control the off-pathway aggregation of $n$ and $m$ species by tweaking the $B_1 \rightarrow B'_1$ bridge.

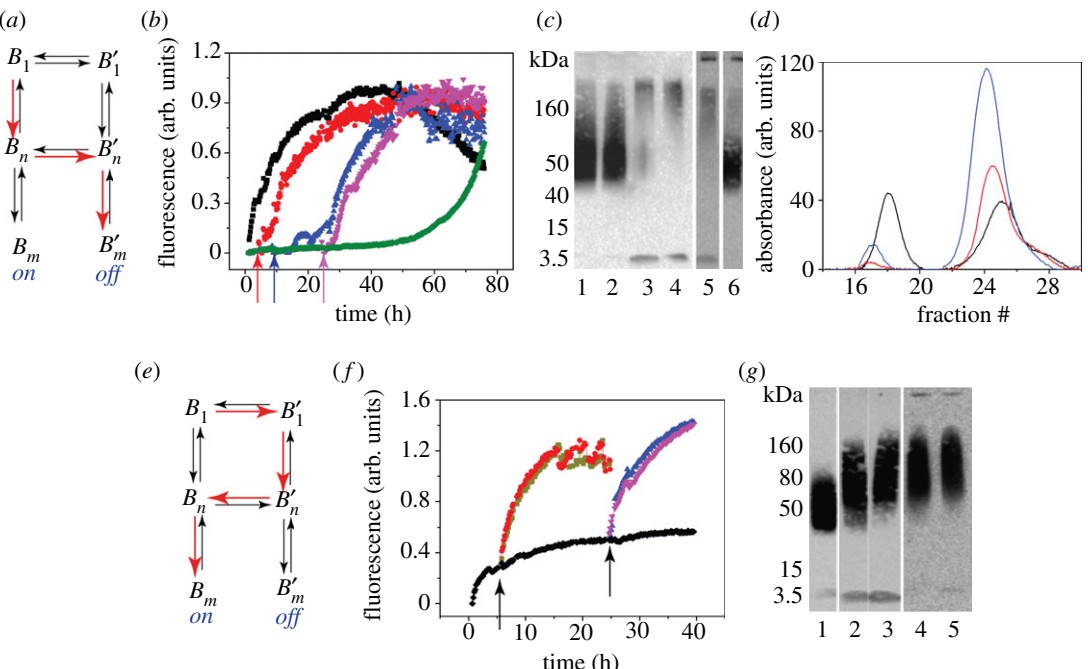

**Figure 5.** Experimental verification of switching of pathways. (a) Schematic representation of on-pathway switching (red arrows) of on-pathway ($B_n$) to off-pathway ($B'_n$ and $B'_m$) on addition of C12 FA. (b) ThT kinetics of the on-to-off transitions probed by the introduction of fatty acid at 3 (red circle), 8 (blue triangle) and 24 h (purple inverted triangle) time points, along with the controls with no fatty acid (green diamond) and with C12 FA introduced at 0 h (black square). (c) Immunoblots for the corresponding reactions: addition of 5 mM at 3 h (lane 1); addition of 5 mM at 8 h (lane 2); 3, 8 and 24 h buffer controls (lanes 3–5), and addition of 5 mM at 24 h (lane 6). (d) SEC fractionation of the reaction before the addition of fatty acid at 24 h (blue), involving the addition of 5 mM C12 fatty acid (black) at 24 h to the sample and control without fatty acid (red), after subsequent incubation for 24 h at 37℃. (e) Schematic representation for switching of off-pathway ($B'_n$) to on-pathway ($B'_n$ to $B_n$) on dilution of the fatty acid below its critical micelle concentration (f). ThT kinetics monitored by the removal of 5 mM fatty acid on the sample incubated with Aβ by diluting with buffer either 5- (red or purple) or 10-fold (green or blue) at 5 h (red circle) and 24 h (blue triangle), respectively. The control without dilutions is shown in black; black diamond. (g) Immunoblot of off-pathway oligomer control generated in the presence of 5 mM fatty acid at 24 h (lane 1); 5- and 10-fold dilutions at 5 h, respectively (lanes 2 and 3), and 5- and 10-fold dilutions at 24 h, respectively (lanes 4 and 5).

## 3.2. Biophysical evidence for the switching of aggregation pathways

The effect of FAs on Aβ42 aggregation has been well established in the Rangachari laboratory [13,44,45]. Specifically, using sodium laurate at concentrations near and well above its CMC, the surfactant was able to modulate Aβ42 aggregation toward off-fibril formation pathway that was populated by low-molecular weight oligomers. At concentrations well below CMC, the fatty acid adopted an on-fibril formation pathway [16]. To experimentally assess the switching of pathways from on- to off-pathway and vice versa by modulating $L$ concentrations, kinetic rate differences in aggregation was investigated using ThT dye.

Switching of on- to off-pathway (depicted schematically in figure 5a) was initiated by the addition of 5 mM C12 FA to 25 μM Aβ42 buffered in 20 mM Tris, 50 mM NaCl at pH 8.0. The addition of C12 FA resulted in an increase in ThT fluorescence without any observable lag time (black square; figure 5b). By contrast, Aβ42 in the absence of C12 FA showed a lag phase of approximately 50 h before an increase in ThT fluorescence was observed (black diamond; figure 5b). This behaviour in the presence of C12 FA has been previously observed to generate 12–24mer oligomers of Aβ along the off-fibril formation pathway [48]. In order to evaluate the propensity of bridging from on- to off-pathway, 5 mM C12 FA was added to the control Aβ42 reaction after 0 h (positive control (black square), 3 h (red circle), 8 h (blue triangle) and 24 h (purple inverted triangle). Each of such incubations resulted in an exponential increase in ThT fluorescence suggesting switching of pathways from on to off (figure 5b). Analysis of these samples was also performed using a partially denaturing gel electrophoresis (low SDS and no boiling) and immunoblotting (figure 5b). Injections of C12 FA at 3 and 8 h show the presence of 48–60 kDa band corresponding to 12mer oligomers (lanes 1 and 2, respectively) as compared with the corresponding controls generated upon adding buffer in place of

C12 FA (lanes 3 and 4), which show monomers and some on-pathway aggregates. This suggests that off-pathway oligomers are generated (figure 5c). Similarly, FA injected after 24 h and its corresponding control are shown in lanes 5 and 6, respectively, which shows even after 24 h, C12 FA is able to induce the formation of oligomers to a certain extent, with clearly observable emergence of some on-pathway fibrils. These results parallel those observed by ThT fluorescence (figure 5b).

To further quantify the extent of bridging, the aggregates generated after the 24 h injection of C12 FA (or buffer for the control) were fractionated by SEC, after an additional 24 h incubation (figure 5d). Prior to fractionation, the samples were centrifuged at 18 000g for 20 min to remove any high molecular weight fibrils, and the supernatant was loaded on to the column. After 24 h, the control in the absence of C12 FA shows a small peak near the void volume at fraction 17 and a monomer peak at fraction 24 (blue; figure 5d). Fractionation of the control reaction at 48 h (after injection of buffer at 24 h) showed a diminished peak at fraction 17 and a reduced monomer peak at fraction 24 (red; figure 5d). The reduction in the monomer peak correlates to being consumed during aggregation. A similar reduction in the aggregate peak between 24 and 48 h can be explained by the fact that many have formed fibrils that are centrifuged out. On the other hand, fractionation of the sample after 48 h with the injection of 5 mM C12 FA at 24 h, showed a larger peak at fraction 18 and a reduced monomer peak at fraction 25 (black; figure 5d). This suggests two possibilities: (i) the unreacted monomers adopt off-pathway upon introduction of C12 FA, and/or (ii) the pre-formed aggregates along the on-pathway are re-routed back through the off-pathway, in other words, switching. More detailed analysis on this is discussed later in the article.

To assess a similar switching of pathways, we performed the off- to on-pathway (schematically depicted in figure 5e) switching again using the established C12 FA kinetics. Incubation of 5 mM C12 FA shows an exponential increase in ThT fluorescence (black diamond; figure 5f). To effect switching of off- to on-pathway after certain time periods, the sample was diluted 5- and 10-fold such that the effective concentration of C12 FA drops to 1 and 0.1 mM, which are well below the CMC of the surfactant. It is well established that well below CMC, A$\beta$ aggregation is augmented [16], and therefore, dilutions of 5 mM C12 FA must induce faster rates of aggregation. When dilutions were introduced, at 5 and 24 h time points (arrows; figure 5f), appropriately blank subtracted data showed an increase in ThT fluorescence as expected for both dilutions suggesting the switching of off- to on-pathway (figure 5f). Partially denaturing gel electrophoresis and immunoblotting further confirmed the switching. The 5- and 10-fold dilutions resulted in an increase in the molecular weight of the aggregates including the formation of fibrils both at 5 and 24 h, respectively (lanes 2–5; figure 5f) as compared with the sample in 5 mM C12 FA (lane 1).

## 3.3. Ensemble kinetic simulation models validate the game-theoretic approach in elucidating the dynamics of competing aggregation pathways

### 3.3.1. Parameter estimation

As mentioned in the Material and methods section, in the EKS model, four on-pathway rate constants (namely, $k_{nu}$, $k_{nu\_}$, $k_{el}$, $k_{el\_}$), 10 off-pathway rate constants (namely, $k_{con}$, $k_{con\_}$, $k_{nuf}$, $k_{nuf\_}$, $k_{el1f}$, $k_{el1f\_}$, $k_{el2f}$, $k_{el2f\_}$, $k_{fagf}$, $k_{fagf\_}$) and two off–on switching rate constants were considered (note that the forward and backward rate constants of switching each oligomer was considered the same, leading to only two switching parameters that need to be estimated, i.e. $k_{swi}$, $k_{swi\_}$). Additionally, one needs to estimate two constants: $p$ (which is simply a mapping constant that distinguishes the contributions of on-pathway oligomers from off-pathway oligomers to the ThT signal intensity) and pseudo-micelle concentration (concentration of the fatty acid near its CMC denoted by $L$). Following our published model in [34], the pseudo-micelle concentration was additionally estimated and not calculated directly from the FA concentration values at the CMC, since precise concentrations of pseudo-micelles are difficult to determine experimentally (only a fraction of total fatty acid concentration) as they are in dynamic equilibrium with other phases of micelle formation. This increased the number of parameters needed to be estimated to 18 from the EKS simulations. The potential complication is mitigated by the fact that our on- and off-pathway rate constants can be estimated separately using the respective control data. This makes it less cumbersome to estimate the remaining four rate constants (i.e. the two off–on switching rate constants $k_{swi}$, $k_{swi\_}$, the mapping constant $p$ and the pseudo-micelle concentration $L$) from this off-on switching dataset by significantly reducing the number of free parameters. A large parameter space from $10^{-6}$ to $10^8$ units with multiples of 10, was swept to estimate the value of each of the two switching

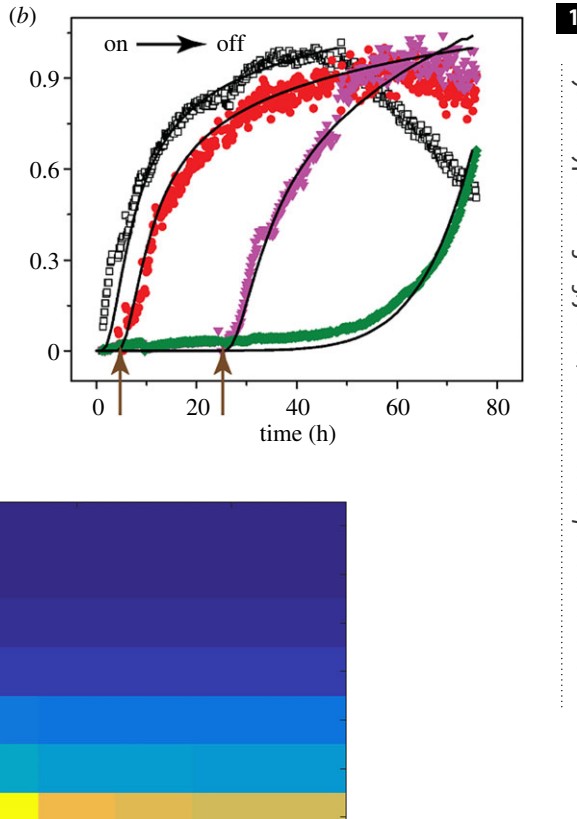

**Figure 6.** Correspondence between experimental results and EKS models on switching of pathways. (*a*) and (*b*) Experimental data (scatter data) on on-to-off and off-to-on pathways reproduced from figure 5*b* and *f*, respectively. Models based on EKS are shown as black lines. The intervention time points of 3 and 24 h (for (*a*)), and 5 and 24 h (for (*b*)) are shown as arrows. Panel (*c*) shows a phase diagram from EKS model at saturation, similar to figure 4 based on variations of the first two bridges. Here, the oligomer ratio of on-pathway to off-pathway was plotted as a heatmap (brighter colour, yellow, denotes on-pathway dominance while darker colour, blue, denotes off-pathway dominance) where the *x*-axis is bridge rate constant $k_{con}$ and *y*-axis is switching rate constant $k_{swi}$. The phase diagram shows a dominance of on-pathway at low values of $k_{swi}$ and $k_{con}$ and dominance of off-pathway for high values of $k_{swi}$ and $k_{con}$.

rate constants. Similarly, the pseudo-micelle concentration was varied from 0.01 to 1 units (with steps of 0.01), and $p$ was varied from $10^5$ to $10^8$ units (with steps of $10^5$). The estimated parameter values corresponding to the best fits are shown in table 2 in Appendix C. The benchmark on- and off-pathway rate constants (estimated separately from control data), were used to estimate the switching rate constants and obtain a global fit to the experimental ThT curves and monomer ratio values estimated from SEC measurements. The average $R^2$ of the off-to-on data is 0.974 and that of on-to-off data is 0.981.

### 3.3.2. Numerical results

The switching rate constants were sensitive specifically in the off–on dataset. The experimental data could not be fit in the absence of the switching rate constants and only a handful of switching rate constant combinations allowed an acceptable fit; the switching rate constants corresponding to the best fit to the experimental data are reported in table 2 in Appendix C. This directly proves the switching of off-pathway oligomers to on-pathway oligomers through the switching pathways due to the dilution of the system. The EKS simulations were conducted in the same way as the experimental set-up. For the off-to-on switching (figure 6), first, combined off- and on-pathway

simulations were executed, up to the switching time-point (of 5 or 24 h); all oligomer concentrations were noted until this point and they were then recalculated based on the amount of dilution at the switching time-point from the experiments. These altered concentration levels for each oligomer were next considered as the initial concentration of the combined off- and on-pathway simulation. Note that the second phase of the off-to-on dataset (figure 6a) did not show any lag time as can be seen in a usual unseeded on-pathway aggregation. Our model predicts a large conversion of off-pathway species to on-pathway oligomers which results in a rapid formation of on-pathway fibrils (denoted by $F$).

For the on–off dataset (figure 6b), stand-alone on-pathway simulations were executed exclusively up to the switching time-point (24 h) and the current oligomer concentrations were noted. These concentrations were then used to restart the combined on- and off-pathway simulation in addition to the pseudo-micelle concentration (that was also estimated in the parameter search step as an independent variable). Surprisingly, we found that the on-to-off-pathway dataset could be fit to our model both considering the switching rate constants, and also in the absence of switching rate constants generating comparable $R^2$ values; in other words, the switching rate constants had low sensitivity to the on–off experimental dataset. Probably as the on-pathway reactions are slow, very little on-pathway oligomers are formed at the switching time-point; as a consequence, this made the switching reaction flux slower than the previous case of off–on switching system resulting in overall lower sensitivity of the switching rate constants to the ThT data points from the experiments. While this precludes precise characterization of on–off switching, we do observe an overall decrease in fibril concentration compared with control data showing at least a qualitative impact of the switching reactions that convert the on-pathway oligomers into off-pathway species. Furthermore, we have also compared the phase diagram of the EKS model by plotting the oligomer ratio of on-pathway to off-pathway (as a heatmap) with varying bridge parameters and switching parameters during the saturation phase (75 h) (figure 6c). The total oligomer count scaled by their size from each pathway was used to compute this ratio. In this heatmap, brighter colour (yellow) denotes a dominance of on-pathway, whereas darker colour (blue) denotes a dominance of off-pathway. By doing so, four phases similar to those obtained from ROM were observed. For a low bridge and switching parameters, a dominance of on-pathway species was observed, whereas for a high value of bridge and switching parameters a prevalent off-pathway was observed (figure 6c); the light yellow and light blue regions depict the mixed pathway zones where both on- and off-pathway oligomers coexist. Note that a one-to-one correspondence between the phase diagrams generated from EKS and ROM models is not possible since the EKS models were built considering a detailed set of reactions, whereas the ROM models correspond to more bulk reactions involving fewer species.

## 4. Discussion

The data presented here is a first attempt in deciphering the complex phenomenon of protein aggregation pathways using a competition-based approach based on classical game theory. Aberrant protein aggregation is sensitive to environmental factors that determine the outcome of the aggregates [38,49]. Using the A$\beta$-fatty acid model system, we have employed a competition-based framework on simplified ROMs to gain preliminary insights. The results re-confirmed our previous observation that fatty acid concentrations modulate A$\beta$ aggregation pathways [34]. Additionally, we discovered that the adoption of on- or off-pathway aggregates tightly depends on a set of rate constant ratios, which in turn suggest the thermodynamic stability (equilibrium constants) of the emerging aggregates. Moreover, $\alpha$ parameters are sensitive to the pseudo-micellar surfactant concentrations, $L$, which hold the key in modulating pathways. The models also provide insights into the feasibility of bridging pathways as a function of emerging higher-order aggregates. For example, the reduced order, six-species model predicts four different scenarios or dominant pathways of reactions which are strongly dependent upon the bridge, while also suggesting that $\alpha_4$ is the key to the formation of larger aggregates in off-pathway. Stability arguments also show the larger aggregates in this system to be more stable (see Appendix B). The EKS simulations display a similar outcome; the simulations indicate that the larger the oligomer, the more significant the impact of that bridge upon formation of the respective fibril.

In our experiments, we note that the propensity to switch pathways is highest when the order of aggregate is the lowest (low-molecular weight) and increasingly becomes weaker as we move toward

higher-order aggregates along either pathway. This is in agreement with the theoretical studies noting the fact that in experiments, high molecular weight species refer to fibrils, while low-molecular weight aggregates then refer to the range of oligomers taken up in EKS and ROM simulations. Perhaps, a significant outcome of this study is the ability of the model to predict the emergence of oligomers by a set of kinetic and thermodynamic parameters, from a 'win' or 'lose' perspective (figure 4). Another key observation is the presence of multiple (neutrally) stable pathways in addition to simplistic on- and off-pathways (figure 4; see Appendix B). The hybrid pathways, especially the off–on domain shown in yellow in figure 4, provide a range of possibilities for $\alpha_3$ and $\alpha_4$ to draw the aggregation dynamics away from toxicity. This is particularly significant for possible intervention strategies, pointing new lines of experimental and theoretical inquiries in the future. Numerical simulations and experiments both clearly support this qualitative result (figures 5 and 6), by revealing dilution to be a clear strategy to force the off–on transition.

Results of ROM indicate that when the ratio of non-bridge forward to backward rates is close to unity, the model achieves equilibrium quickly. However, when forward rates are considerably higher than backward rates, as is to be expected under normal circumstances, the system takes considerably longer to achieve equilibrium due to a cycle of over-shooting of species sizes resulting from a large difference in reaction rates. Similarly, corresponding EKS simulations indicate a $10^{18}$-fold difference in the forward and backward switching rate constants (table 2 in Appendix C) pointing to potentially irreversible effects of switching oligomers between pathways although the system may take a longer time to achieve equilibrium; this observation, however, pertains to our reaction system with fixed initial monomer concentration and is expected to show fluctuating dynamics by considering monomer or pseudo-micelle entry rates and stochastic effects of the switching of oligomers between the pathways.

# 5. Conclusion

The results presented here showcase the applicability of game theory on understanding amyloid aggregation pathways. This is significant because it provides an ability to predict the emergence of aggregates along multiple pathways along a temporal and equilibrium landscape map. Such a map can be further refined to see how it evolves as a function of a given interacting partner of A$\beta$, such as fatty acid as demonstrated here. A significant impact of this work could be realized with the potential for the prediction of the emergence of oligomers, which provides a handle for understanding the conditions at which toxic strains are formed and disappear. The simplified model presented here can be further fine-tuned into more sophisticated models by including more species along pathways, additional pathways and more interacting partners that can modulate the pathway, etc. In sum, the results presented here establish a new paradigm in understanding the complex dynamics of A$\beta$ aggregation and provide impetus towards deciphering amyloid pathogenesis along with making therapeutic and diagnostic advances for such debilitating diseases in the future.

Data accessibility. Data and relevant code for this research work are stored in GitHub: https://github.com/bnetlab/Alzheimer_Ab_switching_ode_model and have been archived within the Zenodo repository: https://doi.org/10.5281/zenodo.3674268.

Authors' contributions. P.G., V.R. and A.V. conceptualized and designed the idea behind the project, and contributed equally in preparing the manuscript. E.S. and A.V. conducted mathematical stability analysis work, P.R. and P.G. conducted simulations, while J.S. and V.R. conducted the biophysical experiments. All authors gave final approval for publication.

Competing interests. The authors declare that they have no competing interests in this work.

Funding. The authors want to thank the National Science Foundation for their financial support; NSF CBET 1802793 (to V.R.), NSF CBET 1802588 (to P.G.) and NSF CBET 1802641 (to A.V.). The authors also thank the National Center for Research Resources (grant no. 5P20RR01647-11) and the National Institute of General Medical Sciences (grant no. 8 P20 GM103476-11) from the National Institutes of Health for funding through INBRE and National Institute of Aging (grant no. R56 AG062292) (to V.R.).

Acknowledgements. The authors also wish to thank Joseph Pateras for his help with some of the calculations.

# Appendix A. Choice of $n$ and $m$ in reduced-order model

The approach taken up in this paper and in our previous research papers as well is to break up the complex problem of protein aggregation to a more tractable and analysable form, via the reduced-order model and also the larger, EKS model which lends itself to the details of the biophysics. The

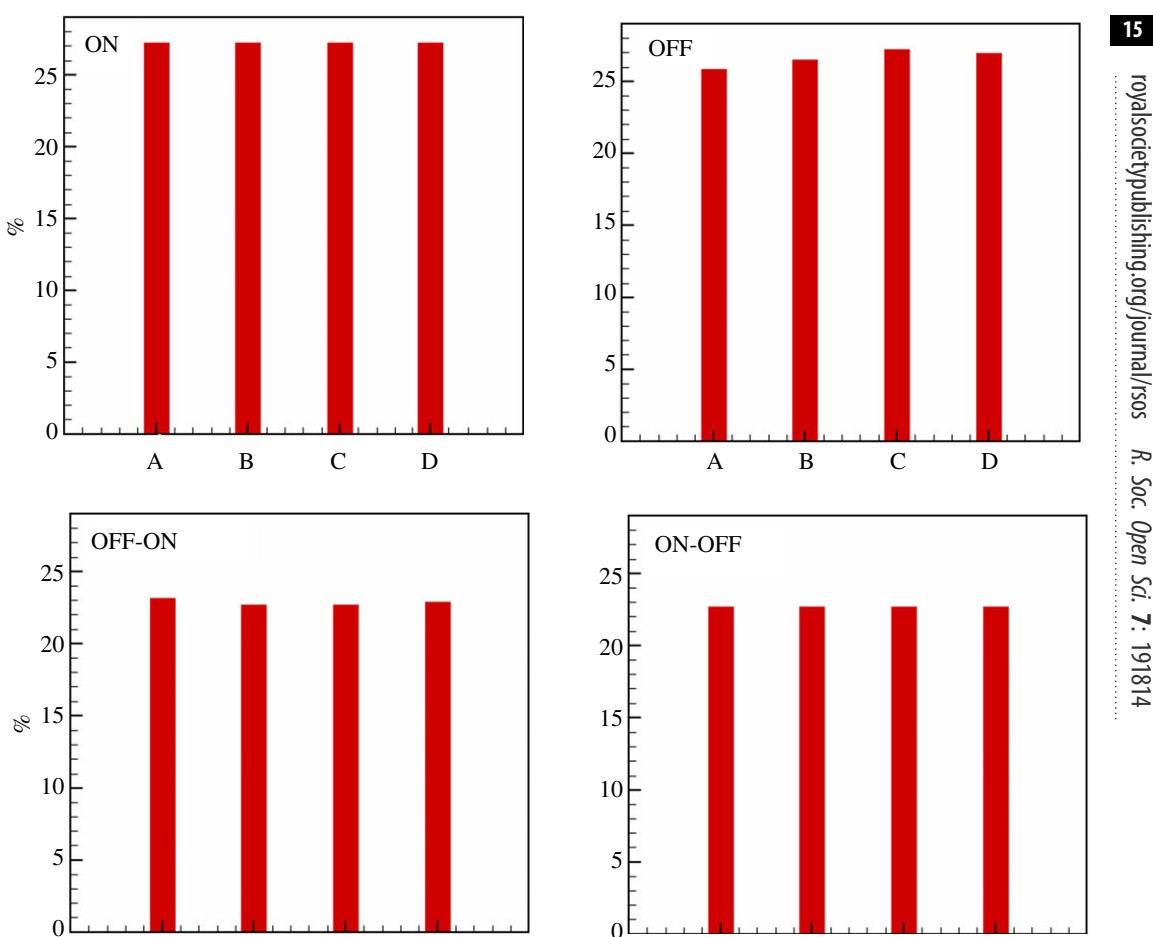

**Figure 7.** Percentage of the phase space occupied by each of the pathways in the six-species system as the choice of $(n, m)$ varies. Note that the different cases in the $x$-axis are to be interpreted as follows: case A: $n = 2$, $m = 4$; case B: $n = 4$, $m = 8$; case C: $n = 8$, $m = 40$ and case D: $n = 12$, $m = 24$.

ROM is, therefore, to be seen as a toy problem, which permits detailed analysis in a manner not possible with the very complex EKS model, where the parameters are all fixed (obtained through curve fitting with the experimental results). The results of the ROM provide insights into the system and help us ask the right kinds of questions about the kinds of experiments and EKS computations that needed to be performed. While some of the figures in the main text are restricted to the case $n = 4$, $m = 20$, these choices are made without loss of generality. Other choices of $n$ and $m$ (figure 7) have also been explored and the outcomes are seen to be qualitatively very similar to the one shown. Figure 7 shows the change in the % of the phase space ($0 < \alpha_3 < 2$, $0 < \alpha_4 < 2$) taken up by each of the four pathways with changes to the pair $(n, m)$. The bar graphs reveal these phases to barely change showing their ubiquity and theoretical significance. Appendix B, which focuses on the stability analysis of the system also reveals similar results.

# Appendix B. Linear stability analysis of reduced-order model

A linear stability analysis was conducted to confirm the conditions under which equilibria are stable and the sensitivity of these solutions to the parameters in this problem. We use the variables $X_1, Y_1, X_n, Y_n, X_m$ and $Y_m$ to represent the concentrations of the various perturbed species, while the equilibria for the monomers and oligomers from the two pathways are indicated by means of an '$e$' in the subscript (i.e. $B_{k,e}$ represents the equilibrium concentration for the oligomer of size $k$). The central idea behind the stability analysis being that a stable equilibrium requires that the perturbed quantities eventually vanish, under certain conditions.

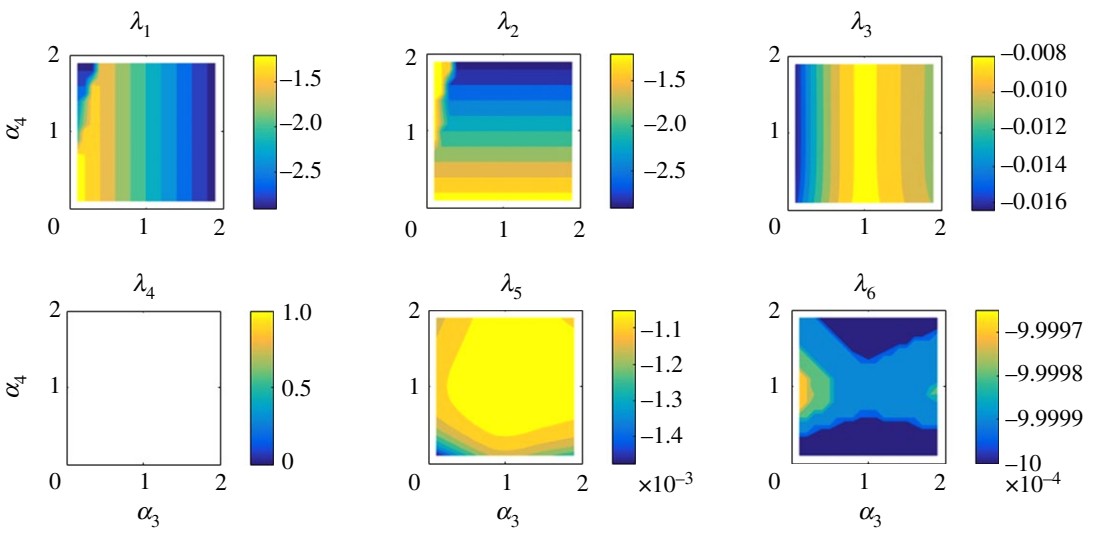

**Figure 8.** $\lambda_i$ as a function of $\alpha_3$ and $\alpha_4$ for the case of $n = 8$, $m = 40$. The deeper blue shade represents the more stable regions. The eigenvalue $\lambda_4$ is 0 for all values of the parameters indicating neutral stability.

The linearized system of equations for the perturbations yields the matrix, given by

$$
\begin{pmatrix}
-(n^2\alpha_2 B_{1,e}^{n-1}+\alpha_3) & 1 & n\alpha_1 & 0 & 0 & 0 \\
\alpha_3 & -(n^2\beta_2 B_{1,e}^{n-1}+1) & 0 & n\beta_1 & 0 & 0 \\
n\alpha_2 B_{1,e}^{n-1} & 0 & -\left(\alpha_1+\alpha_4+\dfrac{m^2}{n}\beta_3 B_{n,e}^{(m/n)-1}\right) & \beta_4 & \dfrac{m}{n}\alpha_5 & 0 \\
0 & n\beta_2 B_{1,e}'^{n-1} & \alpha_4 & -\left(\beta_1+\dfrac{m^2}{n}\alpha_6 B_{n,e}'^{(m/n)-1}+\beta_4\right) & 0 & \dfrac{m}{n}\beta_5 \\
0 & 0 & \dfrac{m}{n}\beta_3 B_{n,e}^{(m/n)-1} & 0 & -\alpha_5 & 0 \\
0 & 0 & 0 & \dfrac{m}{n}\alpha_6 B_{n,e}'^{(m/n)-1} & 0 & -\beta_5
\end{pmatrix}
$$

$$(B\,1)$$

whose eigenvalues (denoted $\lambda_i$, where $i = 1$–$6$) are indicative of the stability of the systems. Of key interest is the effect of the bridge parameters, $\alpha_3$ and $\alpha_4$, and their effect on the stability of each model. A sampling of this effect is captured in figure 8, which depicts the contour plots of the eigenvalues of the *Base Model* for $0 \le \alpha_3, \alpha_4 \le 2$ for the special case when $n = 8$ and $m = 40$. In this figure, the lighter shades depict regions of low stability while the darker ones are more stable. The eigenvalue $\lambda_i$ corresponds to neutral stability. Overall, we find that the stability profile for equilibria corresponding to the *Base Model* does not change much for variations in values of $n$ and $m$. The stability picture for the *Base Model*, however, is significantly different from that of *Model 2*. In the *Base Model*, one of $\lambda_4$ or $\lambda_5$, is always zero for all values of $\alpha_3$ and $\alpha_4$, while for *Model 2*, we observe switching behaviour between $\lambda_2$ and $\lambda_4$, i.e. *Model 2* shows greater sensitivity to the values of the bridge parameters.

We distinguish two different characteristic effects, namely *switching* and *crossing* of the eigenvalues as the two bridge parameters are varied (figure 9). The switching indicates a sudden, drastic change in behaviour of the species, where the course of domination of one species over other is abruptly reversed while the crossing is a more gradual version of this shift. In previous work [34], the switching has been compared with a sort of *transcritical-like bifurcation* in the system. Table 1 shows the switching and crossing points for eigenvalues $\lambda_1$ and $\lambda_2$ as $\alpha_3$ and $\alpha_4$ vary. As can be seen, there is crossing where $\alpha_3 = \alpha_4$, whereas switching has an exponential relationship between the two parameters. A regression model indicates that switching occurs according to $\alpha_4 \approx 2.04 \times 1.55^{\alpha_3}$ with $R^2 = 0.953$.

Simulations for other reaction rate regimes over a larger range of values for $\alpha_3$ and $\alpha_4$ (greater than 2) showed the switching and crossing to persist, as indicated by table 1. In previous studies with lower-dimensional models [34], we have seen such switching to occur as well, which appears to be indicative of the sensitivity of the system to the various pathways and species in the model and activation of one of these pathways under appropriate conditions. For the *Base Model*, with $n = 2$ and $m = 4$, we have $\lambda_3 = 0$ for low $\alpha_3$ and $\alpha_4$, but as we increase these two parameters, $\lambda_2$ vanishes and then finally, with further

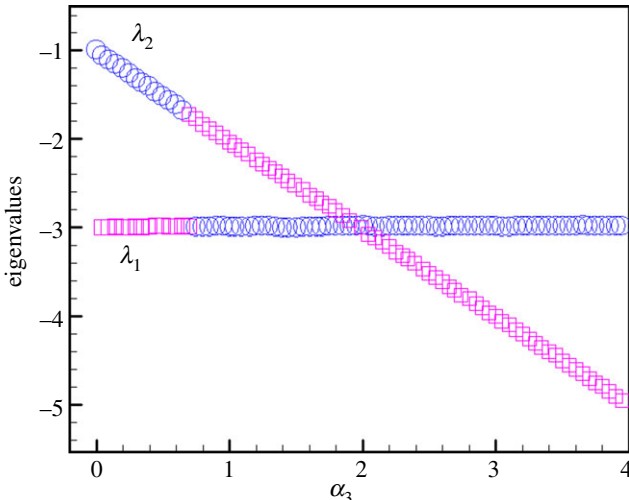

**Figure 9.** A sample case of eigenvalues depicting switching and crossing as a function of the bridge parameter $\alpha_3$. The eigenvalue $\lambda_2$ is denoted in blue while eigenvalue $\lambda_1$ is indicated in lavender.

**Table 1.** The switch and cross points of $\alpha_3$ as a function of $\alpha_4$.

|  | switch | cross |
|---|---|---|
| $\alpha_4$ | $\alpha_3$ | $\alpha_3$ |
| 2 | 0.90 | 2 |
| 4 | 1.53 | 4 |
| 8 | 2.35 | 8 |
| 16 | 4.05 | 16 |
| 32 | 6.65 | 32 |
| 64 | 8.05 | 64 |

increase, $\lambda_4$ goes to zero. Thus, for any rate environment, the stability of the system near the point of equilibrium was found to be neutrally stable for sufficiently large values of the bridge parameters.

The impact of species-size upon stability was also examined by studying the cases of $(n, m)$ equal to $(2, 4)$ and $(8, 40)$ in addition to the standard case $(4, 20)$. Overall, in the case of the *Base Model*, no switching of eigenvalues was observed for a given species size environment as $\alpha_3$ and $\alpha_4$ were varied. For instance, in the *Base Model*, $\lambda_5$ is zero for all values of $\alpha_3$ and $\alpha_4$ when $n = 2$ and $m = 4$, and $\lambda_4$ is zero when $n = 8$ and $m = 40$. However, for *Model 2*, there is switching between $\lambda_2$, $\lambda_3$ and $\lambda_4$ when $n = 2$ and $m = 4$, whereas when $n = 8$ and $m = 40$, we observe switching between $\lambda_1$, $\lambda_2$ and $\lambda_3$. In general, as $n$ and $m$ are increased, the overall magnitude of stability increases, i.e. *the larger the species, the more stable the individual oligomer and also the overall system, appears to be.*

# Appendix C. The ensemble kinetic simulation model

Differential equations of the species:

On-pathway species:

$$\frac{\mathrm{d}A_1}{\mathrm{d}t} = -I_1 - \sum_{i=1}^{11} H_i - H_1,$$

$$\frac{\mathrm{d}A_i}{\mathrm{d}t} = -H_i + H_{(i-1)} - I_i; \quad \forall i \in \{2, 3\},$$

$$\frac{\mathrm{d}A_i}{\mathrm{d}t} = -H_i + H_{(i-1)} - I_i + S_i; \quad \forall i \in \{4, \ldots, 11\}$$

and

$$\frac{\mathrm{d}F}{\mathrm{d}t} = H_{11}.$$

**Table 2.** Estimated parameters from the EKS model.

| parameters | value |
|---|---|
| $k_{nu}$ | $4 \times 10^{-4} \ \mu M^{-1} \ h^{-1}$ |
| $k_{nu\_}$ | $3.5 \times 10^{-2} \ h^{-1}$ |
| $k_{el}$ | $3.5 \times 10^{5} \ \mu M^{-1} \ h^{-1}$ |
| $k_{el\_}$ | $1 \times 10^{-3} \ h^{-1}$ |
| $k_{con}$ | $5 \times 10^{-6} \ \mu M^{-3} \ h^{-1}$ |
| $k_{con\_}$ | $1 \times 10^{-1} \ h^{-1}$ |
| $k_{nuf}$ | $1.4 \times 10^{-1} \ \mu M^{-1} \ h^{-1}$ |
| $k_{nuf\_}$ | $1 \ h^{-1}$ |
| $k_{el1f}$ | $1 \times 10^{4} \ \mu M^{-1} \ h^{-1}$ |
| $k_{el1f\_}$ | $2 \times 10^{-1} \ h^{-1}$ |
| $k_{el2f}$ | $1 \times 10^{3} \ \mu M^{-1} \ h^{-1}$ |
| $k_{el2f\_}$ | $5 \times 10^{-3} \ h^{-1}$ |
| $k_{fagf}$ | $5 \times 10^{3} \ h^{-1}$ |
| $k_{fagf\_}$ | $6 \times 10^{-7} \ \mu M^{-3} \ h^{-1}$ |
| $k_{swi}$ | $5 \times 10^{15} \ h^{-1}$ |
| $k_{swi\_}$ | $1 \times 10^{-3} \ h^{-1}$ |
| $p$ | $2 \times 10^{8} / 8 \times 10^{8}$ |
| $L$ | $50 \ \mu M$ |

Off-pathway species:

$$\frac{dA'_4}{dt} = G'_1 - H'_1 - I'_1 - S_4,$$

$$\frac{dA'_i}{dt} = -H'_i + H'_{(i-1)} - I'_i - S_i; \quad \forall i \in \{5, \ldots, 11\},$$

$$\frac{dF'_1}{dt} = H'_{11} - P'_1 - \sum_{i=1}^{3} P'_i,$$

$$\frac{dF'_i}{dt} = P'_{i-1} - P'_i; \quad \forall i \in \{2, \ldots, 3\},$$

$$\frac{dF'_4}{dt} = P'_3 - R'_1$$

and

$$\frac{dF''_1}{dt} = 4R'_1.$$

Pseudo-micelle

$$\frac{dL}{dt} = -G'_1$$

The simulated signal calculated as follows:

$$signal_{on} = [F] * p,$$

$$signal_{off} = \sum_{i=1}^{4} F'_i; \quad \forall i \in \{1, \ldots, 4\}$$

and

$$signal = signal_{on} + signal_{off}.$$

The estimated parameters are given in table 2.

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
