## [Reviewer comments · Royal Society Open Science]

Review History

RSOS-191814.R0 (Original submission)

Review form: Reviewer 1

Is the manuscript scientifically sound in its present form?

No

Are the interpretations and conclusions justified by the results?

Yes

Is the language acceptable?

Yes

Do you have any ethical concerns with this paper?

No

Have you any concerns about statistical analyses in this paper?

No

Recommendation?

Major revision is needed (please make suggestions in comments)

Comments to the Author(s)

Review: A game theoretic approach to deciphering the dynamics of amyloid-beta aggregation along competing pathways

The authors describe 3 models for the kinetics of amyloid-beta formation, given the presence of certain types of fatty acids. They also propose that two of these models would, at least partially, map onto a game theoretic formalism. The contribution of this paper appears to be as providing evidence for a switching reaction in their chemical kinetics models in addition to proposing how such models could be mapped onto a game theory approach. The former goal is interesting and worthy of publication. While the latter is also interesting and potentially useful, the ideas it encompasses do not go far enough to warrant the status assigned to it by the authors in the abstract and even the title of the paper. Specifically, the authors call the paper "A game theoretic approach..." but said approach is limited to only a proposed payoff function and identification of conditions for Nash equilibrium, but these results are derived from the two models that are not fit to the experimental data. Nevertheless, with some revisions the authors would have a publishable, interesting study on the kinetics of amyloid-beta fibril formation that contributes meaningfully to the existing body of literature. To be clear: I think the content and significance of the work is worthy of publishing, but it has some significant yet tractable issues that the authors should address ahead of publication.

I'll break down my specific concerns into those that are major and minor:

Major concerns:

1) The authors need to give more explanation as to why their "reduced order model" represents kinetics of tetramers only (line 132 specifying $n=4$). To support this modeling picture, the authors invoke a citation (Powers and Powers, 2008), but although Powers and Powers make an illustrative choice of $n=4$, it's not clear why the Ghosh et al. chose to use this specific value. Figure 3 of Powers and Powers (2008) illustrates a plot of the full simulation model vs. this $n=4$ choice. It's clear that $n=4$ captures some dynamics, but it appears to only be an approximation of them. Since this seems to be a central premise of the models in the currently reviewed manuscript, can the Ghosh et al. elaborate on and further justify their specific choice of $n=4$? (The authors DO mention they vary n --on line 296, but only in the context of the effects on the τ_{eq} , and do not mention validity to the full system that I can tell.)

2) The authors indicate how they chose parameters only briefly, around lines 155-157, and in my opinion in an insufficient manner. They indicate that dimensionless rate constants are chosen so that models are consistent with data. However, they go on to use two dimensionless models ("base model" and "model 2"). Are these both consistent with the data despite having seemingly different values for the rate constants? It's not clear to me if both models reflect an arbitrary choice or one supported by data, and if they are supported by data, then how they are extracted from those data. Even if the fit is qualitative, that information should be reported and justified.

3) The authors show that EKS model is able to fit the data pretty well. However, the authors also propose two versions of "reduced order models," which are introduced in the text much earlier, presumably to identify the link between dynamic modeling and a game-theoretic approach. I have two major concerns that explain knowledge gaps the authors need to address:

a) First, the authors should construct a phase diagram of the fitted EKS model and compare it against the one identified in Fig. 4. In the ROMs, the authors illustrate the 4 phases of their model in terms of "bridge parameters" which seem related to the "switching" kinetic rate constants of the EKS approach. So I think they should be able to make such a comparison between long-term steady-states of both modeling approaches.

b) Second, the authors need to compare the dynamics of their ROMs to the dynamics of the fitted EKS model. The fitted EKS model should be the “gold standard” the authors use to assess whether their ROMs are justified, because the EKS model clearly fits the data pretty well. It may be too much to ask here, but I would suggest the authors use the fitted EKS model to identify their parameter values of the ROMs through curve-fitting of the ROMs to either the same dataset as the EKS was fitted, or by matching some equivalent aspects of the ROMs to the EKS model (e.g., the steady-state values).

4) The authors use two ROMs to support the development of a payoff function, which is used in Game Theory to understand competition between decision makers. Although this mapping is interesting, the abstract of the manuscript makes it sound like it is the major point of the article, which it doesn't actually seem to be. In fact, the novelty of the work appears to be in validating the idea of “switching” reactions between on- and off-pathway oligomers, with the game theory connection being an interesting aside. The abstract needs to be adjusted to reflect this.

a) If the authors insist on keeping the connection to game theory at full strength, then at minimum I think they do more to analyze the game theory model. For example, they may want to consider validating the payoff function via production of an EKS phase diagram mentioned in comment 3. Then, it might be worthwhile to analyze the implication of the payoff function with various strategies; although it's not clear to me how biology would “choose” these strategies given that rate constants are fixed through biochemistry.

Minor comments:

5) Please correct inconsistency with notation. For example, fatty acid surfactants are abbreviated Ls on line 96, but then as L in section 2.1.

6) I don't think the authors should refer to “equilibrium states” as the stationary states of their model. It seems as though they are really steady states. I suggest replacing these instances of “equilibrium states” with “steady states,” such as on line 266. As such, re-title section 3.1.1 to something like “steady states” or “fixed points.”

7) What do the authors mean when they say (line 269) “...that concentration of species continue to fluctuate over time for our models...?” Does “fluctuate” mean “oscillate?” If solutions do not converge at long times to a fixed point, then the system would appear to admit limit cycles, which begs a fixed point analysis to understand the types of fixed points allowed by the model and their conditions.

8) On line 287, I think instead of an “approximately equal to” symbol, the authors might actually mean a “proportional to” symbol. Unless, of course, the factor out in front of the r is actually very close to 1.

9) The first time Fig. 4 is mentioned, they refer to panel (D) (line 338). Shouldn't they instead refer to panel (A)? That would mean moving the panels of the figure around so that panels are organized in an order in which they are presented in the text.

10) The authors should address various spelling errors (e.g., see “indicaated” on line 343, line 392 the “and” should not be italicized, line 400 the “to” should not be italicized).

11) The authors should address various grammatical errors (e.g., eliminate comma in line 354).

12) It seems more relevant that Section 3.1.3 be positioned in the Methods section instead of the Results section, given that in the mentioned section, the long-time states of the model system are mapped onto a game-theoretic framework via definition of a payoff matrix.

13) At what density was the parameter space explored on lines 481? It's unclear from the text whether this range was an input into some other algorithm for model fitting purposes or whether they are claiming to have examined values over 14 orders of magnitude, which is an incredible range to sample sufficiently. This absolutely needs clarification.

Review form: Reviewer 2

Is the manuscript scientifically sound in its present form?

No

Are the interpretations and conclusions justified by the results?

Yes

Is the language acceptable?

Yes

Do you have any ethical concerns with this paper?

No

Have you any concerns about statistical analyses in this paper?

No

Recommendation?

Major revision is needed (please make suggestions in comments)

Comments to the Author(s)

The present manuscript deals with amyloid fibres, which are a special secondary structure of proteins. This is an interesting topic. Amyloids have harmful effects in most cases, but can be functional in some others. The authors present a mathematical model of the aggregation of such fibres. Modelling approaches are very useful in this field. In particular, the authors present a game-theoretical approach. A major criticism I have is that the game-theoretical aspects are explained insufficiently and it even remains unclear why such an approach is appropriate here (see my comments below).

Major comments

(1) Repeatedly, the authors use the term "Alzheimer disease". The mostly used English version is "Alzheimer's disease", though.

(2) When discussing the biological relevance, the authors practically exclusively mention "Alzheimer's disease". Only at the end of the Abstract, they briefly say that "oligomer generation [...] is ubiquitously observed in many neurodegenerative diseases." I feel that it is important to discuss this in more detail in the Intro and/or Discussion and to mention examples such as Huntington's disease.

(3) Besides the harmful effects of amyloids, it is also of interest to mention that these structures can play functional roles in some organisms. See Wikipedia entry "Amyloid", section entitled "Non-disease and functional amyloids". One article describing this is D.M. Fowler et al. Functional amyloid formation within mammalian tissue. PLoS Biol. 4 (2006) e6.

(4) Line 94: "we have approached to answer these questions using a game theory-based approach". Here, some Refs. are needed. While the authors cite many Refs. concerning the biochemical background of amyloids, they don't cite any Ref. on game theory in the Intro. As

they apply that theory to a molecular system, it would be appropriate to cite a monograph or review on game theory in general as well as papers dealing with such specific applications. An example paper is:

K. Bohl et al.: Evolutionary game theory: molecules as players. *Mol. Biosystems* 10 (2014) 3066 – 3074.

First ideas on applying game theory to protein binding have been put forward in

I. Kovács et al.: Water and molecular chaperones act as weak links of protein folding networks: Energy landscape and punctuated equilibrium changes point towards a game theory of proteins. *FEBS Lett.* 579 (2005) 579, 2254–2260.

Also worth being cited:

M. Antal et al.: Waves in proteins and protein networks: Applications of percolation and game theories in signaling and drug design. *Curr. Prot. Pept. Sci.* 10 (2009) 161-172

(5) In Section 2.2, some papers on Ensemble Kinetic Simulation are cited. I suggest citing one or two of them already in the Intro, when mentioning that term first.

(6) Lines 126-128: How is the distinction between prime and non-prime species motivated? It is not obvious to me that the off-pathway species are called prime (= first-class, exquisite) and the on-pathway species, non-prime.

(7) The game-theoretical approach followed in the manuscript requires more explanation. What are the players in the game? I assume that the usage of on-pathway and off-pathway are the different strategies, with a third strategy being that both pathways are used to the same extent. In the paper, the term "bridge parameters" is used (for the α_i). I have never seen that term before. In the spirit of game theory, could they be called strategy parameters?

(8) A game-theoretical scenario arises when the payoff (benefit) for one player does not only depend on its own strategy choice but also on the choice of the other player. It remains obscure so far whether the system under study shows such properties. Even more so as it is unclear who the players are. Overall, for the system in question, I do not see the advantage of using a game-theoretical approach. I think that the results could be achieved by a classical, ODE-based approach. The outcome which pathway dominates in certain value ranges of the "bridge parameters" is quite intuitive and by no means surprising.

(9) Line 121: Is the oligomerization a cooperative, all-or-none process? How can it be justified that the dimers and trimers (A2 and A3) can be neglected?

(10) Lines 124, 125: Is the stoichiometric coefficient m/n non-integer? Wouldn't it be better to write the equation as $m A_n \rightleftharpoons n A_m$? This would also circumvent having non-integer exponents in the differential equations, which is unusual.

(11) Lines 283-285: "we found that when the ratio of backward to forward rates is close to 1, the model settles at equilibrium more quickly than when the ratio is large (see figure 2c)." First, the abscissa in Fig. 2c ranges only from 0 to 0.2 and, thus, does not show values close to 1. Second, the label of the abscissa says "Forward to Backward" while in the text, it reads "backward to forward". This may confuse the reader.

(12) The payoff matrix shown in Fig. 4c has unusual entries. Normally, in each position of the matrix, the payoffs for both players are shown. (In the case of symmetric games, it is sufficient to indicate the payoff for one player.) In Fig. 4c, states (e.g. "Mixed", "on = off") are displayed rather than payoffs.

(13) The stability analysis in Appendix 1 is rather cumbersome. In my opinion, the linearized equations (S1)-(S6) are unnecessary. Usually, local stability analysis is performed by writing the Jacobian matrix and calculating its eigenvalues. In fact, that matrix originates from the linearized equations. However, these equations are no longer necessary.

Minor comments

(a) Abstract: "that adopt off-fibril formation pathway" sounds grammatically odd. Can a pathway be adopted? Perhaps better: "that use an off-fibril formation pathway"

(b) Typos: line 45: "strucutral"

l. 64: "strucutre", same line: "morpholgy"

l. 69: "precrsor"

l. 343: "indicaated"

l. 532: „theroetic“

l. 564: „simualtions“

l. 582: "emergnce"

The authors should check again the entire manuscript.

(c) Perhaps, one or two papers by Manuel Than (formerly Jena, Germany) could be cited. He published some work on the amyloid precursor protein, for example:
S.O. Dahms, ..., M.E. Than.: Structure and biochemical analysis of the heparin-induced E1 dimer of the amyloid precursor protein. Proc Natl Acad Sci U S A (2010) 107: 5381-5386.

Decision letter (RSOS-191814.R0)

27-Jan-2020

Dear Dr Ghosh,

The editors assigned to your paper ("A game theoretic approach to deciphering the dynamics of amyloid- β aggregation along competing pathways") have now received comments from reviewers. We would like you to revise your paper in accordance with the referee and Associate Editor suggestions which can be found below (not including confidential reports to the Editor). Please note this decision does not guarantee eventual acceptance.

Please submit a copy of your revised paper before 19-Feb-2020. Please note that the revision deadline will expire at 00.00am on this date. If we do not hear from you within this time then it will be assumed that the paper has been withdrawn. In exceptional circumstances, extensions may be possible if agreed with the Editorial Office in advance. We do not allow multiple rounds of revision so we urge you to make every effort to fully address all of the comments at this stage. If deemed necessary by the Editors, your manuscript will be sent back to one or more of the original reviewers for assessment. If the original reviewers are not available, we may invite new reviewers.

- Data accessibility

If you wish to submit your supporting data or code to Dryad (<http://datadryad.org/>), or modify your current submission to dryad, please use the following link:
<http://datadryad.org/submit?journalID=RSOS&manu=RSOS-191814>

- Competing interests

- Authors' contributions

- Acknowledgements

- Funding statement

on behalf of Professor Samson Abramsky (Associate Editor) and Mark Chaplain (Subject Editor)
 openscience@royalsociety.org

Comments to Author:

Reviewers' Comments to Author:

Reviewer: 1

Comments to the Author(s)

Review: A game theoretic approach to deciphering the dynamics of amyloid-beta aggregation along competing pathways

The authors describe 3 models for the kinetics of amyloid-beta formation, given the presence of certain types of fatty acids. They also propose that two of these models would, at least partially, map onto a game theoretic formalism. The contribution of this paper appears to be as providing evidence for a switching reaction in their chemical kinetics models in addition to proposing how such models could be mapped onto a game theory approach. The former goal is interesting and worthy of publication. While the latter is also interesting and potentially useful, the ideas it encompasses do not go far enough to warrant the status assigned to it by the authors in the abstract and even the title of the paper. Specifically, the authors call the paper "A game theoretic approach..." but said approach is limited to only a proposed payoff function and identification of conditions for Nash equilibrium, but these results are derived from the two models that are not fit to the experimental data. Nevertheless, with some revisions the authors would have a publishable, interesting study on the kinetics of amyloid-beta fibril formation that contributes meaningfully to the existing body of literature. To be clear: I think the content and significance of the work is worthy of publishing, but it has some significant yet tractable issues that the authors should address ahead of publication.

I'll break down my specific concerns into those that are major and minor:

Major concerns:

1) The authors need to give more explanation as to why their "reduced order model" represents kinetics of tetramers only (line 132 specifying $n=4$). To support this modeling picture, the authors invoke a citation (Powers and Powers, 2008), but although Powers and Powers make an illustrative choice of $n=4$, it's not clear why the Ghosh et al. chose to use this specific value. Figure 3 of Powers and Powers (2008) illustrates a plot of the full simulation model vs. this $n=4$ choice. It's clear that $n=4$ captures some dynamics, but it appears to only be an approximation of them. Since this seems to be a central premise of the models in the currently reviewed manuscript, can the Ghosh et al. elaborate on and further justify their specific choice of $n=4$? (The authors DO mention they vary n --on line 296, but only in the context of the effects on the τ_{eq} , and do not mention validity to the full system that I can tell.)

2) The authors indicate how they chose parameters only briefly, around lines 155-157, and in my opinion in an insufficient manner. They indicate that dimensionless rate constants are chosen so

that models are consistent with data. However, they go on to use two dimensionless models (“base model” and “model 2”). Are these both consistent with the data despite having seemingly different values for the rate constants? It’s not clear to me if both models reflect an arbitrary choice or one supported by data, and if they are supported by data, then how they are extracted from those data. Even if the fit is qualitative, that information should be reported and justified.

3) The authors show that EKS model is able to fit the data pretty well. However, the authors also propose two versions of “reduced order models,” which are introduced in the text much earlier, presumably to identify the link between dynamic modeling and a game-theoretic approach. I have two major concerns that explain knowledge gaps the authors need to address:

a) First, the authors should construct a phase diagram of the fitted EKS model and compare it against the one identified in Fig. 4. In the ROMs, the authors illustrate the 4 phases of their model in terms of “bridge parameters” which seem related to the “switching” kinetic rate constants of the EKS approach. So I think they should be able to make such a comparison between long-term steady-states of both modeling approaches.

b) Second, the authors need to compare the dynamics of their ROMs to the dynamics of the fitted EKS model. The fitted EKS model should be the “gold standard” the authors use to assess whether their ROMs are justified, because the EKS model clearly fits the data pretty well. It may be too much to ask here, but I would suggest the authors use the fitted EKS model to identify their parameter values of the ROMs through curve-fitting of the ROMs to either the same dataset as the EKS was fitted, or by matching some equivalent aspects of the ROMs to the EKS model (e.g., the steady-state values).

4) The authors use two ROMs to support the development of a payoff function, which is used in Game Theory to understand competition between decision makers. Although this mapping is interesting, the abstract of the manuscript makes it sound like it is the major point of the article, which it doesn’t actually seem to be. In fact, the novelty of the work appears to be in validating the idea of “switching” reactions between on- and off-pathway oligomers, with the game theory connection being an interesting aside. The abstract needs to be adjusted to reflect this.

a) If the authors insist on keeping the connection to game theory at full strength, then at minimum I think they do more to analyze the game theory model. For example, they may want to consider validating the payoff function via production of an EKS phase diagram mentioned in comment 3. Then, it might be worthwhile to analyze the implication of the payoff function with various strategies; although it’s not clear to me how biology would “choose” these strategies given that rate constants are fixed through biochemistry.

Minor comments:

5) Please correct inconsistency with notation. For example, fatty acid surfactants are abbreviated Ls on line 96, but then as L in section 2.1.

6) I don’t think the authors should refer to “equilibrium states” as the stationary states of their model. It seems as though they are really steady states. I suggest replacing these instances of “equilibrium states” with “steady states,” such as on line 266. As such, re-title section 3.1.1 to something like “steady states” or “fixed points.”

7) What do the authors mean when they say (line 269) “...that concentration of species continue to fluctuate over time for our models...?” Does “fluctuate” mean “oscillate?” If solutions do not converge at long times to a fixed point, then the system would appear to admit limit cycles, which begs a fixed point analysis to understand the types of fixed points allowed by the model and their conditions.

8) On line 287, I think instead of an “approximately equal to” symbol, the authors might actually

mean a “proportional to” symbol. Unless, of course, the factor out in front of the r is actually very close to 1.

9) The first time Fig. 4 is mentioned, they refer to panel (D) (line 338). Shouldn't they instead refer to panel (A)? That would mean moving the panels of the figure around so that panels are organized in an order in which they are presented in the text.

10) The authors should address various spelling errors (e.g., see “indicaated” on line 343, line 392 the “and” should not be italicized, line 400 the “to” should not be italicized).

11) The authors should address various grammatical errors (e.g., eliminate comma in line 354).

12) It seems more relevant that Section 3.1.3 be positioned in the Methods section instead of the Results section, given that in the mentioned section, the long-time states of the model system are mapped onto a game-theoretic framework via definition of a payoff matrix.

13) At what density was the parameter space explored on lines 481? It's unclear from the text whether this range was an input into some other algorithm for model fitting purposes or whether they are claiming to have examined values over 14 orders of magnitude, which is an incredible range to sample sufficiently. This absolutely needs clarification.

Reviewer: 2

Comments to the Author(s)

The present manuscript deals with amyloid fibres, which are a special secondary structure of proteins. This is an interesting topic. Amyloids have harmful effects in most cases, but can be functional in some others. The authors present a mathematical model of the aggregation of such fibres. Modelling approaches are very useful in this field. In particular, the authors present a game-theoretical approach. A major criticism I have is that the game-theoretical aspects are explained insufficiently and it even remains unclear why such an approach is appropriate here (see my comments below).

Major comments

(1) Repeatedly, the authors use the term “Alzheimer disease”. The mostly used English version is “Alzheimer's disease”, though.

(2) When discussing the biological relevance, the authors practically exclusively mention “Alzheimer's disease”. Only at the end of the Abstract, they briefly say that “oligomer generation [...] is ubiquitously observed in many neurodegenerative diseases.” I feel that it is important to discuss this in more detail in the Intro and/or Discussion and to mention examples such as Huntington's disease.

(3) Besides the harmful effects of amyloids, it is also of interest to mention that these structures can play functional roles in some organisms. See Wikipedia entry “Amyloid”, section entitled “Non-disease and functional amyloids”. One article describing this is D.M. Fowler et al. Functional amyloid formation within mammalian tissue. PLoS Biol. 4 (2006) e6.

(4) Line 94: “we have approached to answer these questions using a game theory-based approach”. Here, some Refs. are needed. While the authors cite many Refs. concerning the biochemical background of amyloids, they don't cite any Ref. on game theory in the Intro. As they apply that theory to a molecular system, it would be appropriate to cite a monograph or review on game theory in general as well as papers dealing with such specific applications. An example paper is:

K. Bohl et al.: Evolutionary game theory: molecules as players. *Mol. Biosystems* 10 (2014) 3066 – 3074.

First ideas on applying game theory to protein binding have been put forward in

I. Kovács et al.: Water and molecular chaperones act as weak links of protein folding networks: Energy landscape and punctuated equilibrium changes point towards a game theory of proteins. *FEBS Lett.* 579 (2005) 579, 2254–2260.

Also worth being cited:

M. Antal et al.: Waves in proteins and protein networks: Applications of percolation and game theories in signaling and drug design. *Curr. Prot. Pept. Sci.* 10 (2009) 161-172

(5) In Section 2.2, some papers on Ensemble Kinetic Simulation are cited. I suggest citing one or two of them already in the Intro, when mentioning that term first.

(6) Lines 126-128: How is the distinction between prime and non-prime species motivated? It is not obvious to me that the off-pathway species are called prime (= first-class, exquisite) and the on-pathway species, non-prime.

(7) The game-theoretical approach followed in the manuscript requires more explanation. What are the players in the game? I assume that the usage of on-pathway and off-pathway are the different strategies, with a third strategy being that both pathways are used to the same extent. In the paper, the term "bridge parameters" is used (for the α_i). I have never seen that term before. In the spirit of game theory, could they be called strategy parameters?

(8) A game-theoretical scenario arises when the payoff (benefit) for one player does not only depend on its own strategy choice but also on the choice of the other player. It remains obscure so far whether the system under study shows such properties. Even more so as it is unclear who the players are. Overall, for the system in question, I do not see the advantage of using a game-theoretical approach. I think that the results could be achieved by a classical, ODE-based approach. The outcome which pathway dominates in certain value ranges of the "bridge parameters" is quite intuitive and by no means surprising.

(9) Line 121: Is the oligomerization a cooperative, all-or-none process? How can it be justified that the dimers and trimers (A2 and A3) can be neglected?

(10) Lines 124, 125: Is the stoichiometric coefficient m/n non-integer? Wouldn't it be better to write the equation as $m A_n \rightleftharpoons n A_m$? This would also circumvent having non-integer exponents in the differential equations, which is unusual.

(11) Lines 283-285: "we found that when the ratio of backward to forward rates is close to 1, the model settles at equilibrium more quickly than when the ratio is large (see figure 2c)." First, the abscissa in Fig. 2c ranges only from 0 to 0.2 and, thus, does not show values close to 1. Second, the label of the abscissa says "Forward to Backward" while in the text, it reads "backward to forward". This may confuse the reader.

(12) The payoff matrix shown in Fig. 4c has unusual entries. Normally, in each position of the matrix, the payoffs for both players are shown. (In the case of symmetric games, it is sufficient to indicate the payoff for one player.) In Fig. 4c, states (e.g. "Mixed", "on = off") are displayed rather than payoffs.

(13) The stability analysis in Appendix 1 is rather cumbersome. In my opinion, the linearized equations (S1)-(S6) are unnecessary. Usually, local stability analysis is performed by writing the Jacobian matrix and calculating its eigenvalues. In fact, that matrix originates from the linearized equations. However, these equations are no longer necessary.

Minor comments

(a) Abstract: “that adopt off-fibril formation pathway” sounds grammatically odd. Can a pathway be adopted? Perhaps better: “that use an off-fibril formation pathway”

(b) Typos: line 45: “strucutral”

l. 64: “strucutre”, same line: “morpholgy”

l. 69: “precrsor”

l. 343: “indicaated”

l. 532: „theroetic“

l. 564: „simualtions“

l. 582: “emergnce”

The authors should check again the entire manuscript.

(c) Perhaps, one or two papers by Manuel Than (formerly Jena, Germany) could be cited. He published some work on the amyloid precursor protein, for example:

S.O. Dahms, ..., M.E. Than.: Structure and biochemical analysis of the heparin-induced E1 dimer of the amyloid precursor protein. Proc Natl Acad Sci U S A (2010) 107: 5381-5386.

Author's Response to Decision Letter for (RSOS-191814.R0)

See Appendix A.

RSOS-191814.R1 (Revision)

Review form: Reviewer 1

Is the manuscript scientifically sound in its present form?

Yes

Are the interpretations and conclusions justified by the results?

Yes

Is the language acceptable?

Yes

Do you have any ethical concerns with this paper?

No

Have you any concerns about statistical analyses in this paper?

No

Recommendation?

Accept with minor revision (please list in comments)

Comments to the Author(s)

I encourage the authors to go back through their added text in detail and correct grammatical errors. For example, I think the following line:

L122: "The second model has no known physical basis however, it can be..."

Should be edited to include a semicolon after the word "basis:"

L22: "The second model has no known physical basis; however, it can be

Review form: Reviewer 2

Is the manuscript scientifically sound in its present form?

Yes

Are the interpretations and conclusions justified by the results?

Yes

Is the language acceptable?

Yes

Do you have any ethical concerns with this paper?

No

Have you any concerns about statistical analyses in this paper?

No

Recommendation?

Accept with minor revision (please list in comments)

Comments to the Author(s)

The text can be shortened by at least 5 %. Already the Abstract is quite long.

Here go some suggestions:

Lines 20,21: "We bring forth the use of a framework that is based"  "Our framework is based"

Line 29: Delete "biological"

Line 30: Delete "physiological"

Line 49: Delete "It is now well established that a" (becomes clear from the Refs. in line 52)

Lines 71,72 are somewhat redundant to lines 75-77.

Lines 311,312: Delete " i.e. concentrations that would simultaneously make the right hand sides of equations (1)-(6) vanish for a given set of parameter values"

Minor:

Line 23: "our mathematical models define the dynamics" better "our mathematical models describe the dynamics"

Decision letter (RSOS-191814.R1)

16-Mar-2020

Dear Dr Ghosh:

On behalf of the Editors, I am pleased to inform you that your Manuscript RSOS-191814.R1 entitled "A game theoretic approach to deciphering the dynamics of amyloid- β aggregation

along competing pathways" has been accepted for publication in Royal Society Open Science subject to minor revision in accordance with the referee suggestions. Please find the referees' comments at the end of this email.

The reviewers and Subject Editor have recommended publication, but also suggest some minor revisions to your manuscript. Therefore, I invite you to respond to the comments and revise your manuscript.

- Ethics statement

- Data accessibility

If you wish to submit your supporting data or code to Dryad (<http://datadryad.org/>), or modify your current submission to dryad, please use the following link:
<http://datadryad.org/submit?journalID=RSOS&manu=RSOS-191814.R1>

- Competing interests

- Authors' contributions

- Acknowledgements

- Funding statement

Because the schedule for publication is very tight, it is a condition of publication that you submit the revised version of your manuscript before 25-Mar-2020. Please note that the revision deadline will expire at 00.00am on this date. If you do not think you will be able to meet this date please let me know immediately.

on behalf of Mark Chaplain (Subject Editor)
openscience@royalsociety.org

Associate Editor Comments to Author:

Comments to the Author:

A few remaining grammatical/typographical modifications are suggested, and the Editors would encourage you to take these into consideration and make appropriate changes before submitting a revised paper.

Reviewer comments to Author:

Reviewer: 1

Comments to the Author(s)

I encourage the authors to go back through their added text in detail and correct grammatical errors. For example, I think the following line:

L122: "The second model has no known physical basis however, it can be..."

Should be edited to include a semicolon after the word "basis:"

L22: "The second model has no known physical basis; however, it can be

Reviewer: 2

Comments to the Author(s)

The text can be shortened by at least 5 %. Already the Abstract is quite long.

Here go some suggestions:

Lines 20,21: "We bring forth the use of a framework that is based"  "Our framework is based"

Line 29: Delete "biological"

Line 30: Delete "physiological"

Line 49: Delete "It is now well established that a" (becomes clear from the Refs. in line 52)

Lines 71,72 are somewhat redundant to lines 75-77.

Lines 311,312: Delete " i.e. concentrations that would simultaneously make the right hand sides of equations (1)-(6) vanish for a given set of parameter values"

Minor:

Line 23: "our mathematical models define the dynamics" better "our mathematical models describe the dynamics"

Author's Response to Decision Letter for (RSOS-191814.R1)

See Appendix B.

Decision letter (RSOS-191814.R2)

01-Apr-2020

Dear Dr Ghosh,

It is a pleasure to accept your manuscript entitled "A game theoretic approach to deciphering the dynamics of amyloid- β aggregation along competing pathways" in its current form for publication in Royal Society Open Science. The comments of the reviewer(s) who reviewed your manuscript are included at the foot of this letter.

on behalf of Prof Mark Chaplain (Subject Editor)
openscience@royalsociety.org

Associate Editor Comments to Author:
Thank you for addressing the remaining referees' comments.

Appendix A

We thank the reviewers for their insightful comments which significantly improved the quality of the manuscript. All the major changes in the main manuscript were done in blue. We also improved the presentation of the manuscript by rephrasing several sections; these were however not marked with a change in color. Below, we provide a point by point response to the comments from the reviewers.

Reviewers' Comments to Author:

Reviewer: 1

Major concerns:

1) The authors need to give more explanation as to why their “reduced order model” represents kinetics of tetramers only (line 132 specifying $n=4$). To support this modeling picture, the authors invoke a citation (Powers and Powers, 2008), but although Powers and Powers make an illustrative choice of $n=4$, it's not clear why the Ghosh et al. chose to use this specific value. Figure 3 of Powers and Powers (2008) illustrates a plot of the full simulation model vs. this $n=4$ choice. It's clear that $n=4$ captures some dynamics, but it appears to only be an approximation of them. Since this seems to be a central premise of the models in the currently reviewed manuscript, can the Ghosh et al. elaborate on and further justify their specific choice of $n=4$? (The authors DO mention they vary n --on line 296, but only in the context of the effects on the τ_{eq} , and do not mention validity to the full system that I can tell.)

The approach taken up in this paper and in our previous research papers as well is to break up the complex problem of protein aggregation to a more tractable and analyzable form, via the reduced order model and also the larger, EKS model which lends itself to the details of the biophysics. The ROM is therefore to be seen as a toy problem, in the same spirit intended by Powers & Powers, which permits detailed analysis in a manner not possible with the very complex EKS model, where the parameters are all fixed (obtained through curve fitting with the experimental results). The particular focus of the ROM lies in investigating the significance of the bridges in allowing for various steady states. The referee rightly states that this model only captures the biophysics approximately and we are certainly mindful of that and claim no more. The results of the ROM provide insights into the system and help us ask the right kinds of questions about the kinds of experiments and EKS computations that needed to be performed.

Other choices of n and m (see figure below) have also been explored and the outcomes are seen to be qualitatively very similar to the one shown here so we chose to present a reasonably simple representative case. More specifically for all the different cases analyzed, the phase and stability analysis yield very similar results. This is clearly seen from the figure below which shows the change in the % of the phase space ($0 < \alpha_3 < 2$, $0 < \alpha_4 < 2$) taken up by each of the four pathways with changes to the pair (n,m) . The bar graphs reveal these phases to barely change showing their ubiquity and theoretical significance. Since the focus of the study is to identify the switching and the various emergent phases in the system, we believe that the $n=4$ case can be chosen to represent the qualitative dynamics without loss of generality. This point has now been added to a new Appendix I. We also refer the reviewers to the end of Appendix II where the choice of n,m was also previously discussed in the context of the stability analysis.

The computational advantage of the $n=4$ based computation should also not be discounted; the ROM devoted a lot of time to parametric sweeps of α_3 and α_4 and this choice was also made in the interest of a reasonable time for computational convergence.

Case A: $n=2$, $m=4$; **Case B:** $n=4$, $m=8$; **Case C:** $n=8$, $m=40$; **Case D:** $n=12$, $m=24$

2) The authors indicate how they chose parameters only briefly, around lines 155-157, and in my opinion in an insufficient manner. They indicate that dimensionless rate constants are chosen so that models are consistent with data. However, they go on to use two dimensionless models ("base model" and "model 2"). Are these both consistent with the data despite having seemingly different values for the rate constants? It's not clear to me if both models reflect an arbitrary choice or one supported by data, and if they are supported by data, then how they are extracted from those data. Even if the fit is qualitative, that information should be reported and justified.

The base model in ROM took its parameter values – which are bulk values – guided by the EKS-Experiment fits. The choice of parameters in this problem is a serious issue in itself which is

insufficiently addressed by the curve-fitting techniques adopted by the majority of the researchers in the field. In the absence of high resolution experiments to discern each of the kinetic states, we understand the need to approximate and do the best possible. The ROM *Base Model* is adopted in the same spirit; previous work by the authors have shown that the scale of forward to backward reaction in the system to be a 1000^{th} , which is what we adopt for the pure on- and off-pathway reactions. Since the study focuses on the impact of the bridge, these two parameters are varied.

The *Model 2* is meant as a 'sensitivity study' to show the impact of a slightly different choice of rate constants and could be considered a pathological case. The phase diagrams in figure 4 shows an interesting change in the presence of the various pathways even though the same four phases persist.

3) The authors show that EKS model is able to fit the data pretty well. However, the authors also propose two versions of "reduced order models," which are introduced in the text much earlier, presumably to identify the link between dynamic modeling and a game-theoretic approach. I have two major concerns that explain knowledge gaps the authors need to address:

a) First, the authors should construct a phase diagram of the fitted EKS model and compare it against the one identified in Fig. 4. In the ROMs, the authors illustrate the 4 phases of their model in terms of "bridge parameters" which seem related to the "switching" kinetic rate constants of the EKS approach. So, I think they should be able to make such a comparison between long-term steady-states of both modeling approaches.

Thank you for the suggestion. We have compared the phase diagram of the EKS model by plotting the oligomer ratio of on-pathway to off-pathway (as a heatmap) with varying bridge parameters and switching parameters at the saturation phase (75h) (Figure 6c). The total oligomer count scaled by their size from each pathway was used to compute this ratio. In this heatmap brighter color (yellow) denote a dominance of on-pathway, whereas darker color (blue) denote a dominance of off-pathway. By doing so, four phases similar to those obtained from ROM were observed. For a low bridge and switching parameters, a dominance of on-pathway species was observed, whereas for a high value of bridge and switching parameters a prevalent off-pathway was observed (Figure 6c); the light yellow and light blue regions depict the mixed pathway zones where both on- and off-pathway oligomers coexist. Note that a one to one correspondence between the phase diagrams generated from EKS and ROM models is not possible since the EKS models were built considering a detailed set of reactions, whereas the ROM models correspond to more bulk reactions involving fewer species. The phase diagrams for the EKS model shown below.

Fig: Phase diagram from EKS model at saturation, similar to figure 4 based on variations of the first two bridges. Here, the oligomer ratio of on-pathway to off-pathway was plotted as a heatmap (brighter color, yellow, denote on-pathway dominance while darker color, blue, denote off-pathway dominance) where the x axis is bridge rate constant k_{con} and y axis is switching rate constant k_{swi} . Phase diagram shows a dominance of on-pathway at low values of k_{swi} and k_{con} and dominance of off-pathway for high values of k_{swi} and k_{con} .

We would like to make the following follow-up comment about the overarching philosophy of the modeling approach in this paper in light of this additional evidence provided here. The ROM model makes a conscious choice of the number of species and bridges in this study which permits the convenience of this sort of analysis. The bridge parameters are variables in the system allowing us the opportunity to see the various phases in the system.

In general, for an n -species system with 2 pathways, there are $n^2/2$ possible number of different pathways. Therefore, the EKS system has several number of competing pathways examining which is clearly not possible in a phase diagram, such as the one presented in the paper. A second issue is the presence of the bridges; this paper seeks to ascertain if and at what level the bridges exist. A ROM type model therefore is being used if these bridges can exist and if so at what scales.

The EKS model however has far too many species and bridges and the analysis performed based on the fits to experiments requires that the values of all rate constants be known. The kind of problem that the reviewer is describing is an appropriate approach to take but a highly complex one which is better suited as a follow-up to the current study (and currently being performed). The

complexity of the problem lies in the fact that even if one were to leave open two bridges and used fitted values for all other rate constants, several new questions arise: (a) which two bridges do we choose and (b) why not choose more than just two simultaneous bridges? Etc. The consequence of trying such a thing is that it also increases the number of possible paths (i.e. on, off, on-off, off etc.) by a significant amount so they cannot anymore be represented as a simple phase diagram. Therefore, in making a choice of only a few selective bridges in the EKS model, one runs the risk of skewing the study and providing an incomplete picture. We would say that while this question is a very significant one that absolutely needs to be addressed, it has to be done so systematically. The value of the ROM model lies precisely in serving as an analytical tool to discern the correct structure of the biophysical system. Therefore, such a one-to-one comparison of the two models cannot be made. Above figure based on the EKS model must be taken as an indicator that no matter the size of the system, this competition based analysis has serious merit, is intrinsic to the system and is somewhat independent of the scale of the problem.

b) Second, the authors need to compare the dynamics of their ROMs to the dynamics of the fitted EKS model. The fitted EKS model should be the “gold standard” the authors use to assess whether their ROMs are justified, because the EKS model clearly fits the data pretty well. It may be too much to ask here, but I would suggest the authors use the fitted EKS model to identify their parameter values of the ROMs through curve-fitting of the ROMs to either the same dataset as the EKS was fitted, or by matching some equivalent aspects of the ROMs to the EKS model (e.g., the steady-state values).

Fig: The fit using reduced order model. Green curve denotes the on pathway, pink denote the off-pathway. The dotted point denotes experimental data whereas the solid lines denote ode fit with that data.

To illustrate the difference between ROM and EKS model we have performed ode simulation using the reactions considered in the ROM model. We found the ROM ode model showed a mixed performance to fit the dynamics of on and off pathways. It can fit off-pathway quite well but struggles to fit on-pathway dynamics, especially at longer times. One point to note, in the reduced model we have not considered the reactions involved in the fibril elongation phase since the focus is on the competition of the two pathways at similar scales (i.e. n , m) across both pathways. This can be a reason that the reduced-order ode model is not able to fit on-pathway dynamics. But ROM is still useful to analyze the end state of the system.

We hope that the previous answers also provide the rationale for the ROM and EKS models; the former as an exploratory tool allowing for more rigorous analysis and qualitative overview of the biophysics while the latter is a fixed but more detailed model which allows for a comparison with experiments directly. Having said this, we once again want to caution against over-reliance on simply the curve-fitting approach, when the underlying kinetic framework itself is unclear. This is especially risky in a system, as highly nonlinear as this where one could find several choices of parameters that equally well fit the data.

4) The authors use two ROMs to support the development of a payoff function, which is used in Game Theory to understand competition between decision makers. Although this mapping is interesting, the abstract of the manuscript makes it sound like it is the major point of the article, which it doesn't actually seem to be. In fact, the novelty of the work appears to be in validating the idea of "switching" reactions between on- and off-pathway oligomers, with the game theory connection being an interesting aside. The abstract needs to be adjusted to reflect this.

a) If the authors insist on keeping the connection to game theory at full strength, then at minimum I think they do more to analyze the game theory model. For example, they may want to consider validating the payoff function via production of an EKS phase diagram mentioned in comment 3. Then, it might be worthwhile to analyze the implication of the payoff function with various strategies; although it's not clear to me how biology would "choose" these strategies given that rate constants are fixed through biochemistry.

The referee is right in noticing that we have perhaps been over-enthusiastic about the "game-theoretic" approach in the paper. We still believe that there is an element of novelty in seeing this problem as one of competition between the pathways, we have been hasty in claiming too much of a similarity to conscious game-playing. We have therefore toned down the sections in the paper and abstract to make our viewpoint and approach clearer.

Minor comments:

5) Please correct inconsistency with notation. For example, fatty acid surfactants are abbreviated Ls on line 96, but then as L in section 2.1.

6) I don't think the authors should refer to "equilibrium states" as the stationary states of their model. It seems as though they are really steady states. I suggest replacing these instances of "equilibrium states" with "steady states," such as on line 266. As such, re-title section 3.1.1 to

something like “steady states” or “fixed points.”

Done

7) What do the authors mean when they say (line 269) “...that concentration of species continue to fluctuate over time for our models...?” Does “fluctuate” mean “oscillate?” If solutions do not converge at long times to a fixed point, then the system would appear to admit limit cycles, which begs a fixed point analysis to understand the types of fixed points allowed by the model and their conditions.

The term fluctuation was used improperly and has been corrected now in the revised paper. We simply mean to state the obvious point that numerically the concentrations are still changing and that we set a threshold of 0.1% to select the terminal states in computations. We have looked for and do not see oscillations in the range of parameters explored.

8) On line 287, I think instead of an “approximately equal to” symbol, the authors might actually mean a “proportional to” symbol. Unless, of course, the factor out in front of the r is actually very close to 1.

The reviewer is correct and we have changed that statement accordingly.

9) The first time Fig. 4 is mentioned, they refer to panel (D) (line 338). Shouldn't they instead refer to panel (A)? That would mean moving the panels of the figure around so that panels are organized in an order in which they are presented in the text.

We agree and the text has been suitably edited.

10) The authors should address various spelling errors (e.g., see “indicaated” on line 343, line 392 the “and” should not be italicized, line 400 the “to” should not be italicized).

Done

11) The authors should address various grammatical errors (e.g., eliminate comma in line 354).

Done

12) It seems more relevant that Section 3.1.3 be positioned in the Methods section instead of the Results section, given that in the mentioned section, the long-time states of the model system are mapped onto a game-theoretic framework via definition of a payoff matrix.

We considered doing so earlier but decided to keep it in the results section since the game theoretic framework is after all built upon the ode-system analysis and we found it to be the best

way to tell this story. We have however introduced some text in the methods section introducing this approach.

13) At what density was the parameter space explored on lines 481? It's unclear from the text whether this range was an input into some other algorithm for model fitting purposes or whether they are claiming to have examined values over 14 orders of magnitude, which is an incredible range to sample sufficiently. This absolutely needs clarification.

This is a valid question. In this EKS model, we have four on-pathway rate constants, ten off-pathway rate constants, and two off-on switching rate constants. Additionally, we also need to estimate two additional constants: p (the signal ratio of on to off-pathway oligomers) and pseudo-micelle concentration. Though pseudo-micelle concentration depends on the fatty acid concentration, we did not find any direct method to relate them. So, we opted to estimate it in this EKS simulation itself. That means we need to estimate a total of 18 parameters here. But fortunately, our on-pathway and off-pathway rate constants can be estimated separately using the on-pathway and off-pathway control data respectively. This makes it easier to estimate the only four rate constants from this off-on switching dataset.

We have searched the parameter space from 10^{-6} to 10^8 units with multiples of 10 to estimate the value of the switching rate constants. Next, all parameter values are fine-tuned manually to get a better fit. Similarly, the pseudo-micelle concentration was varied from 0.01 to 1 units (with steps of 0.01), and p was varied from 10^5 to 10^8 units (with steps of 10^5).

.

Reviewer: 2

Major comments

(1) Repeatedly, the authors use the term "Alzheimer disease". The mostly used English version is "Alzheimer's disease", though.

This difference in usage has been in debate for some time and a handful of scientific organizations have started adopting more correct, 'Alzheimer disease' terminology. American Society for Biochemistry and Molecular Biology (ASBMB) specifically has adopted this more than a decade ago and since then we have also been using this terminology in all our publications.

(2) When discussing the biological relevance, the authors practically exclusively mention "Alzheimer's disease". Only at the end of the Abstract, they briefly say that "oligomer generation [...] is ubiquitously observed in many neurodegenerative diseases." I feel that it is important to discuss this in more detail in the Intro and/or Discussion and to mention examples such as Huntington's disease.

Although we agree with the reviewer on the implications of the work in other neurodegenerative diseases, the modeling and premise is based on the work specifically with A β . Despite the biochemical commonality of amyloid formation, every amyloid protein may adopt multiple pathways under conditions that are much different from those of A β . Therefore, we do not believe it is necessary to widen the discussion to include other pathologies.

(3) Besides the harmful effects of amyloids, it is also of interest to mention that these structures can play functional roles in some organisms. See Wikipedia entry "Amyloid", section entitled "Non-disease and functional amyloids". One article describing this is D.M. Fowler et al. Functional amyloid formation within mammalian tissue. PLoS Biol. 4 (2006) e6.

The reviewer is correct in pointing out that the amyloids are not only pathogenic but also are functional. Many functional amyloids have emerged in the last decade that are very interesting. Having said that, we feel that the manuscript is focused on pathways of A β which is a pathogenic protein involved in AD. We also feel that focused introduction for the problem investigated will be important to improve the brevity of the document. Therefore, we feel inclusion of functional amyloids, although important, seems out of place in this manuscript.

(4) Line 94: "we have approached to answer these questions using a game theory-based approach". Here, some Refs. are needed. While the authors cite many Refs. concerning the biochemical background of amyloids, they don't cite any Ref. on game theory in the Intro. As they apply that theory to a molecular system, it would be appropriate to cite a monograph or review on game theory in general as well as papers dealing with such specific applications. An example paper is:

K. Bohl et al.: Evolutionary game theory: molecules as players. Mol. Biosystems 10 (2014) 3066 – 3074.

First ideas on applying game theory to protein binding have been put forward in

I. Kovács et al.: Water and molecular chaperones act as weak links of protein folding networks: Energy landscape and punctuated equilibrium changes point towards a game theory of proteins. FEBS Lett. 579 (2005) 579, 2254–2260.

Also worth being cited:

M. Antal et al.: Waves in proteins and protein networks: Applications of percolation and game theories in signaling and drug design. Curr. Prot. Pept. Sci. 10 (2009) 161-172

Some appropriate references are certainly needed. We have included some of the relevant citations in the revised version of the paper.

(5) In Section 2.2, some papers on Ensemble Kinetic Simulation are cited. I suggest citing one or two of them already in the Intro, when mentioning that term first.

Appropriate changes have been made.

(6) Lines 126-128: How is the distinction between prime and non-prime species motivated? It is

not obvious to me that the off-pathway species are called prime (= first-class, exquisite) and the on-pathway species, non-prime.

Aggregation of A β along multiple pathways is not novel but has been known for quite some time. Our motivation to the present work originates from Kumar et al, 2011, *Plos One* with a follow-up paper from Rana et al, 2018 Sci Reports in which the Rangachari lab has demonstrated that by modulating the concentrations of fatty acid and its concomitant phase transitions, A β aggregation can be modulated. Specifically, at concentrations just below the critical micelle concentrations, A β forms 12-24mers that are kinetically trapped with longer half-lives along a off-fibril formation pathway. We have cited this paper in the manuscript and set the context of prime and non-primed pathway nomenclature.

(7) The game-theoretical approach followed in the manuscript requires more explanation. What are the players in the game? I assume that the usage of on-pathway and off-pathway are the different strategies, with a third strategy being that both pathways are used to the same extent. In the paper, the term "bridge parameters" is used (for the α_i). I have never seen that term before. In the spirit of game theory, could they be called strategy parameters?

As requested, we have added more to the manuscript to explain the connections to game-theory a bit more clearly. We believe there is an advantage and an element of novelty in seeing this problem as one of competition between the pathways. The readers should try not to make a one-one connection to elements of game theory as employed in more standard works; some of the language employed here (such as 'bridge parameters') is unique to this work and we would prefer to maintain it as such in light of the larger discussions in the paper. However we have also made clarifications in the text to identify these parameters as 'strategy parameters'

The document has been appropriately revised to clarify the approach taken in this paper a bit more clearly.

(8) A game-theoretical scenario arises when the payoff (benefit) for one player does not only depend on its own strategy choice but also on the choice of the other player. It remains obscure so far whether the system under study shows such properties. Even more so as it is unclear who the players are. Overall, for the system in question, I do not see the advantage of using a game-theoretical approach. I think that the results could be achieved by a classical, ODE-based approach. The outcome which pathway dominates in certain value ranges of the "bridge parameters" is quite intuitive and by no means surprising.

The approach being used to describe the competition between the various species is indeed based upon a classical system of nonlinear ODEs. As mentioned earlier, the connection to game-theory is more in a qualitative sense than through quantitative standard methods; it appears to be an effective way to tell our story. That the system should display switching from on to the off pathway is perhaps 'obvious' as the reviewer indicates but the detailed mechanics of the switching is certainly not something to be expected. The existence and conditions under which the hybrid pathways (namely the ones we label 'off-on' and 'on-off') is definitely somewhat unexpected. In a system as nonlinear as this, we would argue that intuition is insufficient. What is perhaps being

overlooked is that there are eight parameters in the system and the outcome of the dynamics is made easier to understand and predict based on the non-dimensionalization that is chosen.

(9) Line 121: Is the oligomerization a cooperative, all-or-none process? How can it be justified that the dimers and trimers (A2 and A3) can be neglected?

This is definitely a cooperative process; all species maintain a non-zero steady state over the entire range of the parameters that are investigated. The phase diagram (figure 4) simply indicates the dominant pathways in the system for a fixed choice of α_3 and α_4 . In each state however, all paths exist simultaneously. This is an important point and has now been made clear in the paper. Figure 3 shows the ratio of the unprimed and primed species as a function of α_3 , α_4 and indicate a smooth transition with increase in these parameters. We can therefore infer that at each stage, the dominance of one of the species does not preclude the existence of the others.

The EKS model treats dimers and trimers as well but the ROM model focuses only on a small set of species (6 in total). The philosophy of the ROM approach is to qualitative capture the essential features of the protein aggregation process in a tractable manner using a bulk modeling approach. It therefore cannot account for all species in the real system (dimers, trimers etc). The ROM analysis provides the subsequent for the EKS model which does account for all intermediate species as well. The validity of these approaches is clear from the predictions made and the experimental verification which reveals the presence of different pathways of aggregation.

(10) Lines 124, 125: Is the stoichiometric coefficient m/n non-integer? Wouldn't it be better to write the equation as $m A_n \leftrightarrow n A_m$? This would also circumvent having non-integer exponents in the differential equations, which is unusual.

We do not really see any conflict in writing the coefficient as m/n . It is understood that m is an integral multiple of n , which also makes physical sense. The revised text explains this point better.

(11) Lines 283-285: "we found that when the ratio of backward to forward rates is close to 1, the model settles at equilibrium more quickly than when the ratio is large (see figure 2c)." First, the abscissa in Fig. 2c ranges only from 0 to 0.2 and, thus, does not show values close to 1. Second, the label of the abscissa says "Forward to Backward" while in the text, it reads "backward to forward". This may confuse the reader.

We thank the reviewer for this observation. For sake of clarity we have decided to replace Figure 2c with the following table which provides just enough information to make our point clear.

Time to Equilibrium	
Backward/Forward Ratio	Equilibrium Time
0.0001	5,190
0.001	2,200
0.01	595
0.1	105
1	20

The text has been correct to align with the label used here.

(12) The payoff matrix shown in Fig. 4c has unusual entries. Normally, in each position of the matrix, the payoffs for both players are shown. (In the case of symmetric games, it is sufficient to indicate the payoff for one player.) In Fig. 4c, states (e.g. “Mixed”, “on = off”) are displayed rather than payoffs.

The payoff matrix is used in a qualitative sense and in this particular case, cannot be assigned specific values since each cell represents the ‘dominant path’ whose values are highly dependent on various parameters. What matters here is which network pathway wins under each restrictive condition. The matrix as presented is meant to be generally valid, no matter the specific parameter values chosen for analysis and must be read as such.

(13) The stability analysis in Appendix 1 is rather cumbersome. In my opinion, the linearized equations (S1)-(S6) are unnecessary. Usually, local stability analysis is performed by writing the Jacobian matrix and calculating its eigenvalues. In fact, that matrix originates from the linearized equations. However, these equations are no longer necessary.

This is a good suggestion. The equations have now been changed by the appropriate matrix.

Minor comments

(a) Abstract: “that adopt off-fibril formation pathway” sounds grammatically odd. Can a pathway be adopted? Perhaps better: “that use an off-fibril formation pathway”

Done.

(b) Typos: line 45: “strucutral”

l. 64: “strucutre”, same line: “morpholgy”

l. 69: “precrsor”

l. 343: “indicaated”

l. 532: „theroetic“

l. 564: „simualtions“

l. 582: “emergnce”

The authors should check again the entire manuscript.

Done.

(c) Perhaps, one or two papers by Manuel Than (formerly Jena, Germany) could be cited. He published some work on the amyloid precursor protein, for example:

S.O. Dahms, ..., M.E. Than.: Structure and biochemical analysis of the heparin-induced E1 dimer of the amyloid precursor protein. Proc Natl Acad Sci U S A (2010) 107: 5381-5386.

Done.

Appendix B

We thank the reviewers for their suggestions and for closely reading the manuscript. All the minor changes suggested by the reviewers have now been incorporated. We also improved the presentation of the manuscript by rephrasing several other sections and taking care of the grammatical errors; these were however not marked with a change in color. Below, we provide a point by point response to the comments from the reviewers.

Reviewers' Comments to Author:

Reviewer: 1

I encourage the authors to go back through their added text in detail and correct grammatical errors. For example, I think the following line:

L122: "The second model has no known physical basis however, it can be..."

Should be edited to include a semicolon after the word "basis:"

L22: "The second model has no known physical basis; however, it can be

Done. We have additionally proof-read the entire manuscript and took care of other grammatical errors.

Reviewer: 2

The text can be shortened by at least 5 %. Already the Abstract is quite long.

Here go some suggestions:

Lines 20,21: "We bring forth the use of a framework that is based"  "Our framework is based"

Line 29: Delete "biological"

Line 30: Delete "physiological"

Line 49: Delete "It is now well established that a" (becomes clear from the Refs. in line 52)

Lines 71,72 are somewhat redundant to lines 75-77.

Lines 311,312: Delete "

i.e. concentrations that would simultaneously make the right hand sides of equations (1)-(6) vanish for a given set of parameter values"

All of these changes were made. We have additionally reduced other portions of the text and altered some of the references.

Minor:

Line 23: "our mathematical models define the dynamics" better

"our mathematical models describe the dynamics"

Done.